# ALIGNING BRAINS INTO A SHARED SPACE IMPROVES THEIR ALIGNMENT TO LARGE LANGUAGE MODEL

## ABSTRACT

Large language models (LLMs) have been shown to perform remarkably well in predicting human neural activity measured using electrocorticography (ECoG) during natural language comprehension. Contextual embeddings from LLMs are typically evaluated against brain activity using linear encoding models estimated at each electrode within each subject. In most studies, these electrodes are simply combined across subjects and encoding models are not evaluated in terms of whether they generalize across subjects. In this paper, we analyze neural responses in 8 subjects while they listened to the same 30 minute podcast episode. We use a shared response model (SRM) to estimate a shared information space across subjects and show that LLM-based encoding models achieve significantly better performance in predicting the shared information features than the original brain responses. We also show that we can use this shared space to denoise the individual brain responses by projecting back to the neural space and this process achieves a mean 38% improvement in encoding performance. More detailed inspection of this improvement in different brain areas reveals that the improvements are the most prominent in brain areas specialized for language comprehension, specifically in superior temporal gyrus (STG) and inferior frontal gyrus (IFG). Finally, we show that the shared space calculated from a group of subjects generalizes to a new subject, allowing us to build linguistic encoding models that generalize across subjects.

## 1    INTRODUCTION

Large language models (LLM) trained using the self-supervised objective of predicting the next word in the context of preceding words perform very well on natural language tasks, such as novel text generation, translation, summarization, and question-answering (Lewis et al. (2019). Liu et al. (2019), Kenton & Toutanova (2019), Yang et al. (2019), Bengio et al. (2000), Radford et al. (2018), Radford et al. (2019), Brown et al. (2005), Zaheer et al. (2020)). Recent studies in neurolinguistics have begun to explore LLMs as computational cognitive models of human brain activity during context-rich, real-world language processing (Nastase et al. (2020a)).  Recent work has used linear encoding models to predict human brain activity measured using fMRI and or ECoG from the contextual embeddings—the hidden states of the LLM, where both the LLM and the human receive the same input (Antonello et al. (2023), Schrimpf et al. (2021), Goldstein et al. (2022b), Caucheteux & King (2022)). The high spatiotemporal resolution of invasive ECoG recordings in particular has begun to provide finer-grained insights into the representational structures and processes shared between LLMs and the brain (Goldstein et al. (2022b), Goldstein et al. (2022a), Goldstein et al. (2023a), Goldstein et al. (2023b), Zada et al. (2023)).

When humans listen to the same natural language stimulus (e.g. a spoken story), our neural activity keys into shared features of stimulus, ranging from low-level acoustic features to higher-level linguistic and narrative features (Nastase et al. (2019)). Although there is coarse-grained alignment across brains, finer-grained cortical topographies for language representation are highly idiosyncratic across individuals (Fedorenko et al. (2010), Lipkin et al. (2022)). Hyperalignment methods have been developed in the fMRI literature to resolve these topographic idiosyncrasies and aggregate multi-subject brain data into a shared information space (Haxby et al. (2011), Chen et al. (2015), Haxby et al. (2020)). Unlike fMRI, where there is already voxelwise feature correspondence, ECoG presents a more difficult correspondence problem because each subject has a different number of

electrodes in different locations; how to best aggregate electrodes across patients is a matter of ongoing research (e.g. Owen et al. (2020)). For this reason, encoding models are typically estimated within individual subjects and are not evaluated in terms of their generalization to new subjects (e.g. Huth et al. (2016), Schrimpf et al. (2021), Goldstein et al. (2022b)).

In this paper, we measured the neural responses of eight ECoG subjects implanted with invasive intracranial electrodes while they listened to a natural language stimulus—i.e. a podcast. We use a shared response model (SRM) (Chen et al. (2015)) to aggregate the subjects' neural data and find the features of the stimulus shared across the participants. In parallel, we use LLMs to extract contextual embeddings for each word of the podcast. The shared informative features and contextual embeddings are then used to build an encoding model. We show that the SRM yields significantly higher encoding performance than the original individual-specific electrodes. Moreover, we show that we can use this shared space to denoise individual subject responses by projecting from the shared space back into the individual neural space. Furthermore, we study encoding performance for different brain areas and find that the SRM-based denoising is giving the highest improvement in brain areas specialized for language comprehension. Finally, we show that the SRM allows us to construct encoding models that generalize across subjects.

## 2 MATERIALS AND METHOD

### 2.1 DATA COLLECTION AND PROCESSING

We recorded the neural activity of eight participants (4 females, 20-48 years) using ECoG while they listened to a 30-minute audio podcast "So a Monkey and a Horse Walk Into a Bar: Act One, Monkey in the Middle" taken from the 'This American Life' podcast. We manually transcribed the story and aligned it to the audio by labeling the onset and offset of each word. An independent listener manually evaluated the alignment. In the podcast, there were 5013 words. For each word, a 1600-dimensional contextual embedding was extracted from the final layer of the GPT-2-XL (Radford et al. (2019)) large language model. These embeddings were reduced to 50 dimensions using principal component analysis (PCA) for the core encoding analyses, based on Goldstein et al. (2022b).

For ECoG data collection, there were a total of 917 electrodes placed on the left hemisphere and 233 on the right hemisphere. The data is sampled at 512 Hz and the line noise harmonics are excluded. We use a band-pass filter to extract the high gamma range 70-200 Hz of the signal. Based on a significance test, we retained 184 electrodes (150 in the left hemisphere, 34 in the right hemisphere) across subjects for further analysis. A detailed description of the electrode number and their coverage across brain areas is given in the Table 4 in appendix. We also incorporated a behavioral experiment where 50 healthy adult subjects (different than the subjects ECoG data collected from) predicted each upcoming word given the context of previous words. Once participants made their predictions, we revealed the actual word and asked them to predict the next word. The process was repeated for all words in the podcast, where for each word, we gathered 50 predictions. We computed the predictability score for each word as the number of subjects who correctly predicted that word divided by the total number of subjects. A predictability score of 1 means all subjects correctly predicted the word, and a score of 0 means no subject could guess the word.

### 2.2 ELECTRODE-WISE ENCODING MODEL OVER TIME

We use contextual word embeddings to predict held-out neural data for individual electrodes. Hence, we build an encoding model using ordinary least-squares multiple linear regression that uses these embeddings to predict neural responses for each electrode at varying lags relative to word onset. At each lag, we take a 200 ms window and compute an average. We employ a 10-fold cross-validation procedure to assess the performance of these models in predicting neural responses for held-out data. For the performance metric, Pearson correlation coefficient was computed between the predicted and the actual signal. This procedure has been repeated for 161 lags from -2000 ms to 2000 ms in 25 ms increments relative to word onset.

In case of comparing the encoding performance of two cases we perform paired t-tests between the two correlation scores across folds for each lag. To correct for testing across multiple lags, we use False Discovery Rate (FDR) (Benjamini & Hochberg (1995)) correction. Lags with p-values less than 0.01 are considered to be significant.

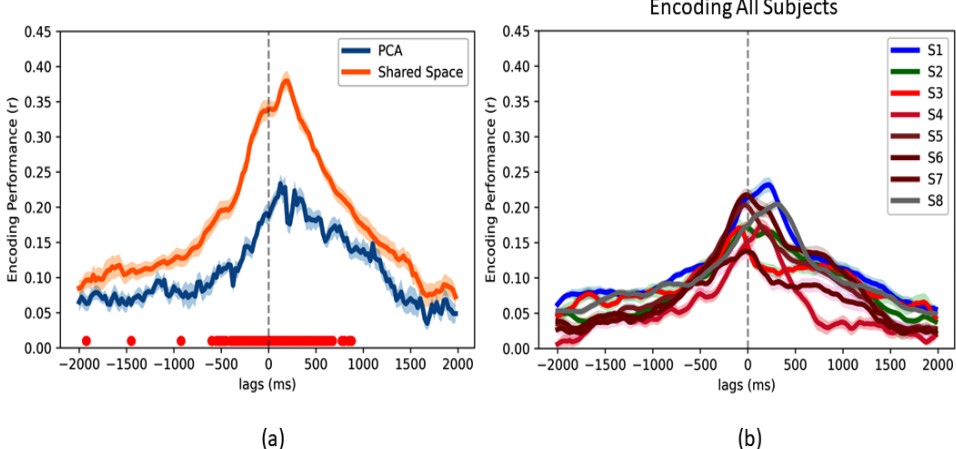

Figure 1: (a) Encoding in Shared Space and PCA reduced neural data. The red horizontal dots indicate the significant lags (p < 0.01, FDR corrected). (b) Encoding performance for each subject's original neural data.

## 2.3 SHARED RESPONSE MODEL

For the podcast dataset, although all the subjects listen to the same story, their corresponding neural activities differ. Hence, we use a shared response model (SRM) (Chen et al. (2015)) to aggregate multi-subject ECoG data into a common space that accounts for different functional topographies. SRM learns subject-specific mappings to a shared space using training data, and then uses these learned mappings to convert each subject's idiosyncratic neural activity into the shared space.

To make this more explicit, let $\{X_i \in \mathbb{R}^{e \times d}\}_{i=1}^m$ be the training data (e electrodes over d time points) for m subjects. We use this data to learn subject-specific bases $W_i \in \mathbb{R}^{e \times k}$ (where k is a hyperparameter that corresponds to the number of components in the new, shared space) and a shared matrix $S \in \mathbb{R}^{k \times d}$, such that $X_i = W_i S + E_i$, where $E_i$ is an error term corresponding to the subject's original brain response. The bases $W_i$ can be thought of as representing the individual functional topographies and $S$ as a latent feature that captures a component of the response that is shared across subjects. In order for the solution to be unique, $W_i$ is subject to the constraint of linearly independent columns and as like (Chen et al. (2015)), $W_i$ is assumed to have orthonormal columns, $W_i^T W_i = I_k$. The following optimization problem is solved to estimate $W_i$ and the shared response $S$-

$$\min_{W_i, S} \sum_i \|X_i - W_i S\|_F^2$$
$$s.t. \ W_i^T W_i = I_k \tag{1}$$

The S and W parameters of the SRM model are jointly estimated using a constrained EM algorithm. We can utilize the learned subject-specific bases to project data into the individual shared response subspace ($S_i$) and denoise our original data $X_i$, such that

$$S_i = W_i^T X_i$$
$$\hat{X}_i = W_i S_i \tag{2}$$
$$\hat{X}_i = W_i W_i^T X_i$$

## 3 EXPERIMENTS

### 3.1 LINGUISTIC ENCODING IN SHARED SPACE

A shared response model (SRM) is fitted on the training data of 8 subjects using hyperparameter K=5, and the shared space and the corresponding subject-specific weights have been learned. First, we fit an encoding model to predict the shared neural activity of all subjects as represented by the 5 components in the shared space, from contextual embeddings extracted from GPT-2-XL. We carry out this analysis for lags from 2000 ms before word onset to 2000 ms after with a 25 ms step, fitting a separate encoding model per lag. For encoding model evaluation, we use ten fold cross-validation. At each lag, we use SRM to learn the subject-specific weights ($W_i$) and construct the shared space ($S_{train}$) using only the training folds. Using these weights, we compute the shared space for the test fold by calculating the averaged shared response across subjects, i.e., $S_{test} = 1/m \sum_{i=1}^{m} W_i^T X_i^{test}$. We use encoding analysis to predict the shared space features $S_{test}$ from the contextual embeddings. Figure 1(a) shows the encoding performance. To evaluate how well the shared space learned by SRM is, we run two additional models for comparison. First, we carry out the same encoding analysis on each individual subject's original neural data at the same lags (Figure 1(b)). Second, as SRM is a form of dimensionality reduction, we employ principal component analysis (PCA) as a control. We project each subject individually to K components, same as SRM shared space, then average each component across subjects. Then we carry out an encoding analysis on it (Figure 1(a)). In Figure 1(a), we see that shared space achieves significantly higher encoding performance than PCA (p < 0.01, FDR corrected). This control analysis show that the stronger encoding performance is not due to the decreased dimensionality of the shared space. Also comparing with Figure 1(b), we see that shared space encoding performs better than the original neural data.

### 3.2 DENOISING NEURAL ACTIVITY USING SHARED SPACE

We want to see whether projecting a subject's neural activity into the shared space and back (i.e., to denoise it) will increase encoding model performance for that subject. For this, we use the subject-specific weights to reconstruct their SRM denoised data as shown in Eq.equation 2. First, we calculate the individual subject's shared subspace $S_i$, by multiplying with the learned subject-specific weights and then project it back to the subject's neural space. Then, we perform an encoding analysis for each subject using the SRM-denoised data and compare it with the encoding performances using the subjects' original neural data. Figure 2 shows the electrode location for each subject, as well as their average encoding performance for both the original and the SRM-denoised neural data. We find that the SRM-denoised data significantly improve encoding performance for all subjects (p < 0.01, FDR corrected), with a 38% average increase in correlation scores.

### 3.3 GENERALIZING SHARED SPACE TO NEW SUBJECT

Since all subjects heard the same story, the information content regarding the stimulus across subjects should be similar. As SRM computes the shared aspects of the multi-subject brain responses, it is expected to generalize to a new subject. To test this hypothesis, we want to see how encoding analysis performs on the shared space estimated for new subjects unseen during SRM training. In this case, SRM training data does not include the neural data of the subject of interest. The learned shared space ($S$) is used as a template, and we calculate the subject-specific weights $W_j$ for the new subject $j$, using the training data $X_j$. To achieve this, we minimize the mean squared error $min_{W_j, W_j^T W_j = I_k} \|X_j - W_j S\|_F^2$ to find $W_j$. Now we explore the generalizability of SRM in both the shared space and the neural space. For the first case, we compute the shared subspace for the test fold of subject $j$ using $S_{test}^j = W_j^T X_{test}^j$. Then we build an encoding model trained on the embeddings of the training fold and the shared space $S$ calculated from SRM training, and try to predict $S_{test}$. We carry out this process for each lag for all the subjects. In Fig. 3, we show the mean performance across subjects for shared space generalization. For comparison, we also show shared space encoding performance where SRM is trained on all subjects' data, and the mean encoding performance of original neural data. It can be seen from the figure that when SRM is trained with all the data, the performance is highly superior, which is expected. But in case of subject's omission, the shared space provides better encoding performance (33% improvement) than the subject's encoding performance with their original neural responses.

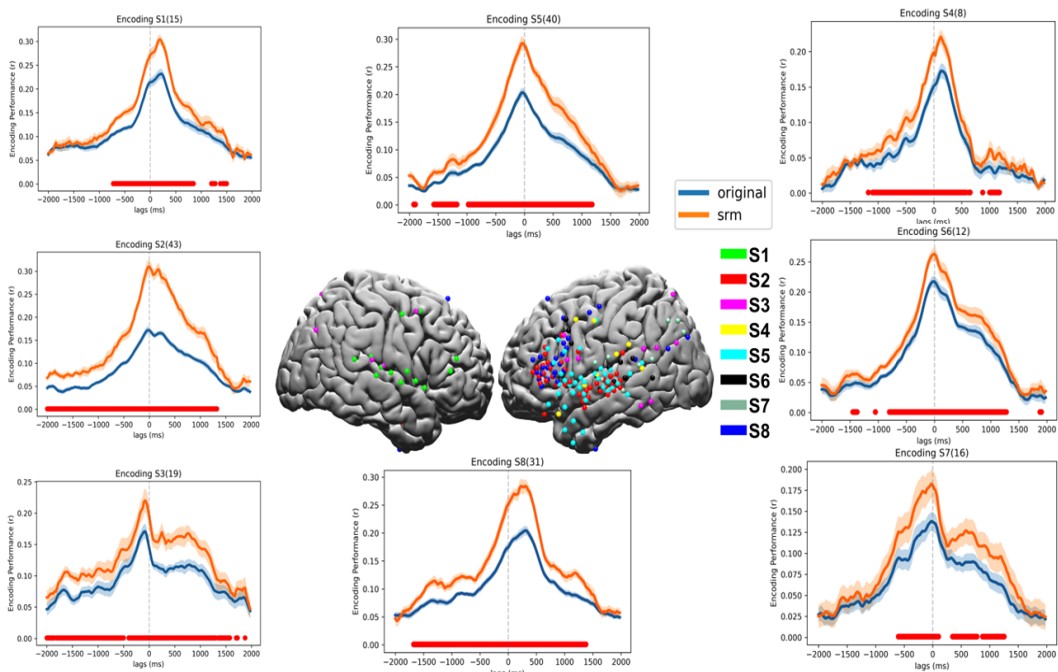

Figure 2: Electrode placements of all the subjects and the encoding performance of all the subjects for original and SRM-denoised brain responses. The red horizontal line indicates the significant lags (p < 0.01, FDR corrected)

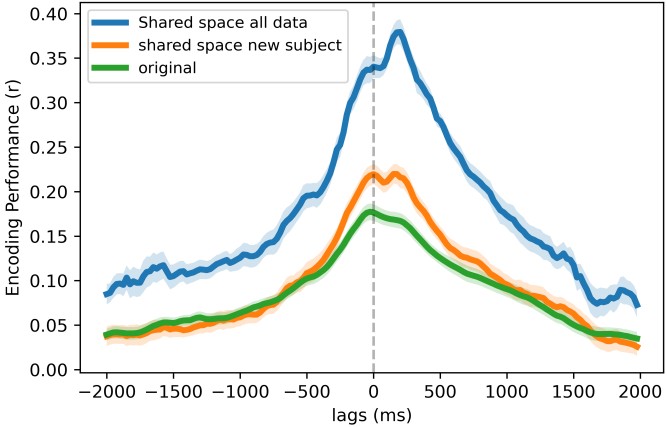

Figure 3: Encoding Performance in three cases- (i) shared space of SRM, where the shared space is calculated using all the subjects training data, (ii) shared space generalized to subject j, where the data of j is not present during SRM training. The process is repeated for all subjects j and the mean result is reported, (iii) original neural data of the subjects.

In order to examine the generalizability of SRM in the individual neural space instead of the shared space, we use the estimated $W_j$ to denoise both train and test data (as shown in equation 2) for new subject $j$, and perform the encoding analysis. We carry out this process for each subject respectively, and Fig. 6 in appendix shows the final result. We see that even though SRM was not trained on the subject's data, by using the shared space calculated from training as a template, we can still estimate subject-specific weights ($W_j$) and use it to denoise the original neural data. It achieves similar higher encoding performance as we got in Fig. 2. Both analyses show that SRM can generalize to new subjects and the shared space calculated among a group of subjects is generalizable to a new subject

as long as the subject experiences the same stimulus.

## 3.4 ANALYZING ENCODING IMPROVEMENTS WITH SRM DENOISING ACROSS BRAIN AREAS

In order to uncover which brain regions improve the most, we quantify the amount of improvement in encoding model performance for each electrode separately. In Fig. 4 we report the improvement for all the electrodes of all the subjects where the improvement is calculated as the difference of encoding correlation achieved using the SRM-denoised data and the original data. The figure illustrates that the highest improvements occur mainly in the inferior frontal gyrus (IFG) and the superior temporal gyrus (STG). We also report the statistics for this improvement showing the number of electrodes for different ranges of improvement in Table 1. Out of 184 electrodes, with SRM-denoising, 168 electrodes have showed improvements, where the maximum absolute improvement has been 0.33. Furthermore, we show the encoding performance of SRM-denoised data and the corresponding original neural data for different brain areas across the subjects in Fig. 5. From the figure, we can observe that across all areas, SRM-denoising yields better performance compared to the original data. We further investigate STG by separating it into three sub-areas- aSTG, mSTG and cSTG, and we find that significantly higher encoding correlation are found at the IFG, aSTG and mSTG area (p < 0.001, FDR corrected), whereas aSTG is registering the highest improvement. We also notice that cSTG, angular gyrus (AG) and temporal lobe (TP) areas are showing small improvement and do not have significant lags. One reason may be the low electrode numbers in these areas.

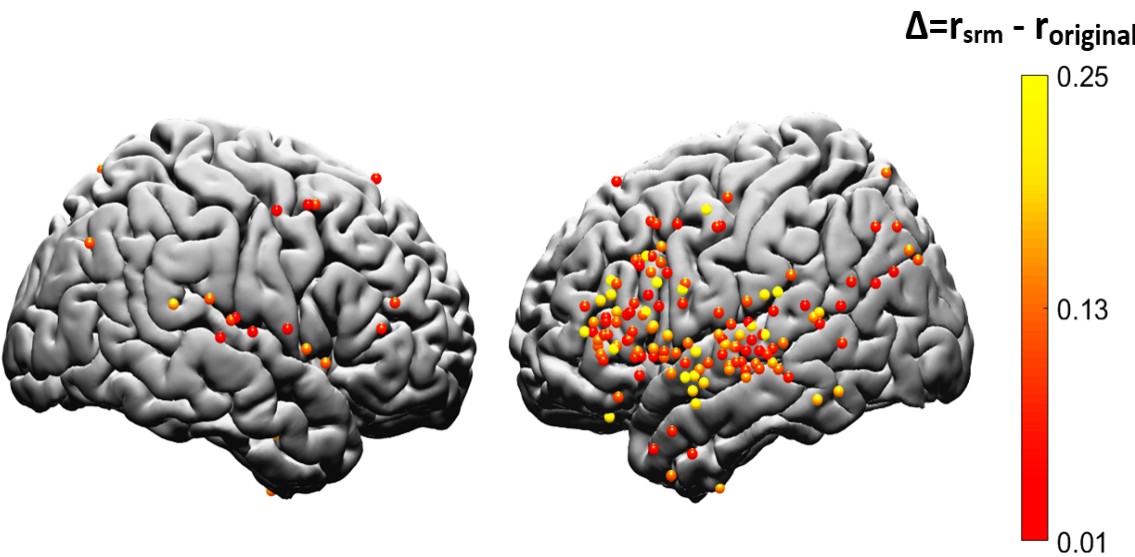

Figure 4: Electrode-wise improvement in encoding performance using SRM-denoised data

Table 1: Statistics for Electrode wise improvement

| Range | number of electrodes |
|---|---|
| >0.2 | 24 |
| 0.15-0.2 | 12 |
| 0.1-0.15 | 43 |
| 0.05-0.1 | 50 |
| 0.01-0.05 | 39 |

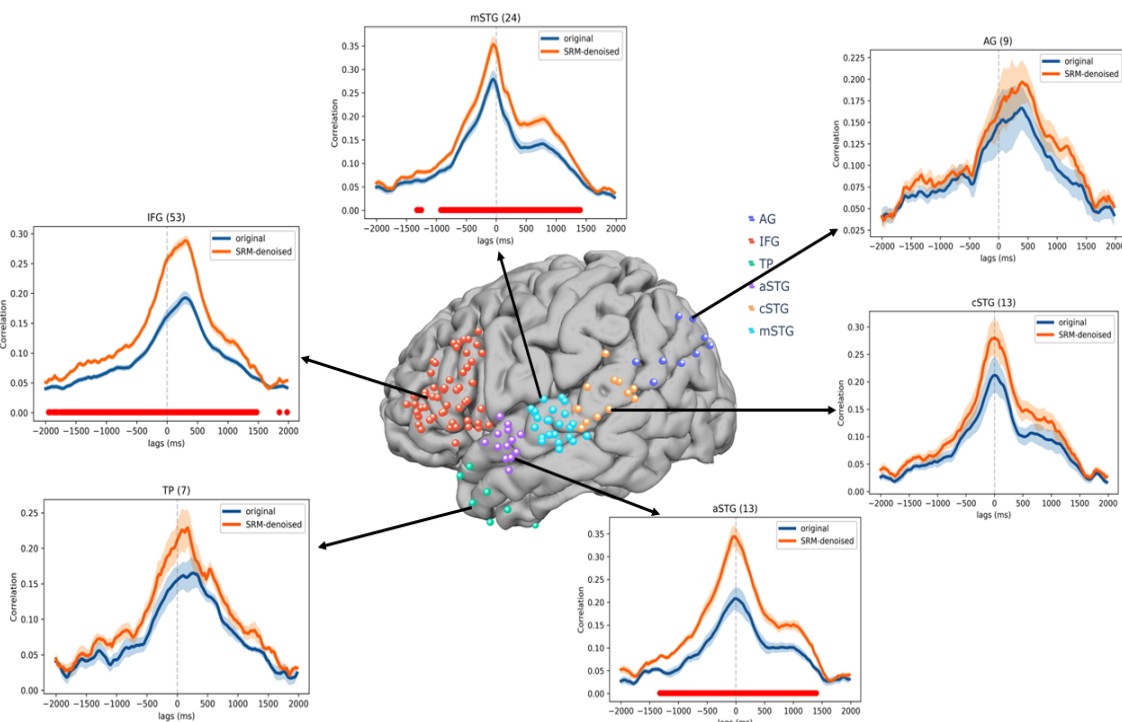

Figure 5: Encoding Performance of original neural data and SRM denoised neural data for different brain areas. The red horizontal line indicates the significant lags (p < 0.01, FDR corrected).

## 3.5 SHARING INFORMATION ACROSS SUBJECTS AND ACROSS REGIONS

In order to establish a measure of shared information across subjects (without reference to any particular encoding model), we design an experiment where we reconstruct the neural signal for a left-out subject j on the left-out segment of the stimulus using the data of other subjects, and quantify how well this correlates with the actual neural signal for that test subject j and test stimulus. Higher correlation indicates more shared information across subjects. To elaborate, first, we train a SRM model on the training data for all the subjects except $j$ using eq. 1. Then using the training data $X_j$ for subject $j$, we find the mapping matrix $W_j$ by minimizing the mean squared error of $min_{W_j, W_j^T W_j = I_k} \|X_j - W_j S\|_F^2$.

Next, we estimate the average shared response of test data for all the subjects except $j$ using $S_{test} = 1/m \sum_{i \neq j} W_i^T X_i^{test}$. With the estimated shared response for the test data, we reconstruct the test data for subject $j$ by $X_{test}^r = W_j S_{test}$. Finally, we calculate the correlation for all the electrodes between $X_{test}^r$ and $X_{test}^j$. We repeat this process for each subject for all the test folds at the word onset. Table 2 summarizes the result. From the Table, we see that SRM-based reconstruction from other subjects achieves 0.25 correlation on average. That is, we can reconstruct one subject's neural activity from other subjects' neural activity at a 0.25 correlation; this is one formulation of a noise ceiling.

We want to compare this SRM-based noise ceiling to a simpler noise ceiling used in prior work. However, we do not have multiple presentations of the same test stimulus from the same subject, and thus must rely on other subjects to estimate the noise ceiling. Furthermore, we do not have electrode correspondence across ECoG subjects (unlike voxels in fMRI). To circumvent these challenges, for a given test subject j, we obtain the electrode-wise signals for a set of test words and find the maximum correlation with all other electrodes across all other subjects. We compute the mean of these maximum intersubject correlations across electrodes and across subjects, and report the results in Table 2. From the table, we see that SRM yields a higher, more robust noise ceiling across subjects.

Table 2: Sharing information across subjects (Values denote average correlation across electrodes ± standard error across cross-validation test sets)

| Subject id | Correlation between the original test data and the reconstructed data | Noise Ceiling |
|---|---|---|
| S1 | 0.28 ± 0.008 | 0.12± 0.005 |
| S2 | 0.21± 0.006 | 0.1± 0.002 |
| S3 | 0.21± 0.012 | 0.1± 0.005 |
| S4 | 0.23± 0.011 | 0.09± 0.005 |
| S5 | 0.29± 0.008 | 0.13± 0.004 |
| S6 | 0.32± 0.009 | 0.13± 0.004 |
| S7 | 0.2± 0.009 | 0.11± 0.008 |
| S8 | 0.23± 0.007 | 0.1± 0.004 |
| **average** | **0.25± 0.009** | **0.11± 0.005** |

Next, we want to explore whether the amount of shared information across subjects differs across brain regions. We re-run the above reconstruction analysis separately for two different brain regions: superior temporal gyrus (STG) and inferior temporal gyrus (IFG). For this analysis, we only considered subjects with five or more electrodes in the region of interest. Table 3 indicates better reconstruction across subjects in STG than in IFG, but both benefit from SRM at a similar magnitude.

Table 3: Area wise information sharing

| Electrode Location | Correlation between the original test data and the reconstructed data | Noise Ceiling |
|---|---|---|
| All electrodes | 0.25 | 0.11 |
| IFG | 0.16 | 0.08 |
| STG | 0.32 | 0.21 |

## 4    DISCUSSION

There are multiple studies that build encoding models to predict neural responses from the contextual embeddings of LLMs (Goldstein et al. (2022b), Goldstein et al. (2023c)). In this paper, we show that aligning brains into a shared space significantly improves the encoding performance. It suggests that the shared space allows LLMs to capture the neural basis of language comprehension better. The shared space captures stimulus features, the information that is preserved across brains, that are not related to one particular individual (Nastase et al. (2020b), Chen et al. (2015)). Moreover, the shared response has been shown to correlate with narrative understanding (Nastase et al. (2019)). Our results show that this shared space is more aligned with LLM embeddings than any individual subjects, and more so than generic PCA (Figure 1). Moreover, our analysis shows that this shared space can generalize to new subjects—i.e., we project a held-out subject into the shared space and show that their encoding model performance increases. This suggests that the LLM model can be used as a shared linguistic model for how information is shared across brains.

We introduce an analysis where we quantify how well each test subject's neural activity can be reconstructed from other subjects. This gives a measure of shared information across subjects and also this reconstruction quality serves as a between-subject "noise ceiling" by resolving idiosyncrasies in feature spaces across subjects. Our analysis shows that the SRM reconstruction across subjects yields considerably higher correlations than naive noise ceiling estimates based electrode-wise intersubject correlations Nastase et al. (2019). We further explore the performance improvement gained by using SRM as a denoising method. By denoising idiosyncratic neural data through the shared space, we find improvement for most of the electrodes (168 out of 184, details in Table 1), with the highest improvement in the brain's language network, namely the IFG and STG area. With

further division in STG, we find that aSTG area is registering the highest improvement. The higher level of the brain area (aSTG) is showing more improvement than the lower levels (mSTG, cSTG). One reason may be that the shared space may be of higher level features related to semantic and understanding of the whole story instead of lower level acoustics features. The most improvement at language related areas suggests that the brain alignment is associated with the semantic information coded in the brain. This observation of brain aligning on the semantic space as it is captured by autoregressive deep language models., supports the new and exciting field of using deep learning as a computational framework for cognitive neuroscience (Richards et al. (2019)).

We also compare the contextual word embeddings derived from GPT2-XL to a baseline of static, non-contextual embeddings derived from GloVe (Pennington et al. (2014); appendix section A.7). We find that the contextual embeddings outperform the static embeddings. Importantly, we also see that the SRM improves encoding performance for both types of embeddings. Furthermore, we compare encoding performance across layers of GPT2-XL (appendix section A.4) and find that intermediate layers provide the best performance. Importantly, we also find that SRM improves encoding performance across all the layers. We also compare encoding performance for several different sizes of LLM (appendix section A.5). We observe a weak relationship suggesting that encoding performance may improve for larger models (Antonello et al. (2023)); importantly, however, we see that SRM improves performance across all models.

In the literature, there are multiple studies using the shared response model (SRM) for fMRI data (Nastase et al. (2020b), Chen et al. (2015), Finn et al. (2020), Cohen et al. (2017)). Unlike fMRI, where there is already voxelwise feature correspondence, ECoG presents a more difficult correspondence problem because each subject has a different number of electrodes in different locations. In most ECoG research, such as Goldstein et al. (2022b), electrodes are simply combined across subjects to construct a "supersubject". No mapping from one subject to another is made—and, critically, whether encoding models actually generalize across subjects has not been investigated. How to best aggregate electrodes across subjects is a matter of ongoing research (e.g. Owen et al. (2020)). Applying SRM to better aggregate electrode signals across brains, and showing how this benefits encoding model performance paves a new avenue for future research. SRM capturing stimulus property across brains and generalizing to new subjects can have potential applications in brain signal decoding (Metzger et al. (2023), Willett et al. (2023)), brain-computer-interface (Sliwowski (2023)).

## 5 Conclusion

Large Language Models (LLM) have provided a unified computational modeling framework for studying the neural basis of human language where we can predict brain signals from the contextual embeddings of the models. In this paper, we seek to understand how different subjects comprehend the same stimulus. We employ a shared response model (SRM) to compute a shared space and the subject specific mappings. The shared space captures the information shared across brains. We extract embeddings from LLM for all the words in the stimulus and run encoding models for both shared space and original neural data, separately. We see a dramatic improvement in encoding performance for the shared space features which demonstrates that the LLMs are capturing the core information content in the brain better compared to predicting the original neural response which includes subject-specific local properties. We also show the effectiveness of the shared space by using it to denoise individual neural responses. Using the SRM-denoised data, we achieve a mean 38% improvement in encoding performance across the subjects. We introduce an analysis where we quantify how well each test subject's neural activity can be reconstructed from other subjects shared space and this gives a measure of shared information across subjects. Our analysis shows that SRM-based reconstruction from other subjects achieves a 0.25 correlation on average. This is one formulation of a noise ceiling. Our exhaustive experiments show that the shared information space learned by SRM is generalizable to new subjects and provides better encoding performance in both the shared and the neural space. Furthermore, our examination of encoding performance for different brain areas tells us that SRM's shared space-based denoising achieves the highest improvement in the brain areas specialized for language comprehension- superior temporal gyrus (STG) and inferior frontal gyrus (IFG). This paves the way for a better scientific understanding of language processing across multiple brains with shared stimulus and suggests that the LLM can be used as a linguistic model for shared information across brains.

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

# A APPENDIX

## A.1 ELECTRODE SELECTION

Phase of each of the electrodes signal was randomized preserving the autocorrelation of the signal to disconnect the relationship between the brain signal and the words, and performed encoding for each electrode. This whole process was repeated for 5000 times where in each step, for each electrode, the maximum values of encoding models across all lags had been stored. This gave us a distribution of 5000 values that was used to estimate significance. We also calculated the maximum encoding performance across lags with non permuted electrode signal and computed the p-value using this value as the percentile from the null distribution of previously computed 5000 values. FDR was used to correct for multiple electrodes. If the $q$ value of the electrode was less than 0.01, it was regarded as significant.

## A.2 SUBJECT WISE ELECTRODE LIST

Table 4: Subject wise Electrode List for different brain areas

| Subjects | Right Hemisphere | Left Hemisphere | | | | | | | | | | | | |
|---|---|---|---|---|---|---|---|---|---|---|---|---|---|---|
| | | IFG | STG | AG | TP | Precentral | Parietal | postcg | aMTG | pMTG | premotor | MFG | other | total |
| S1 | 15 | | | | | | | | | | | | | 15 |
| S2 | | 18 | 16 | | | 1 | 1 | | 1 | | | | 6 | 43 |
| S3 | 10 | 2 | | 3 | | 1 | | | | 2 | | | 1 | 19 |
| S4 | | | 3 | | 1 | 1 | 2 | | | | 1 | | | 8 |
| S5 | 2 | 11 | 18 | | 5 | 2 | | | 1 | | | | 1 | 40 |
| S6 | | 3 | 6 | | | | 1 | | | 1 | | 1 | | 12 |
| S7 | 5 | | 3 | 4 | | | 1 | 1 | | 1 | | | 1 | 16 |
| S8 | 2 | 19 | 1 | 2 | 1 | 1 | | | | | 1 | 3 | 1 | 31 |
| | | | | | | | | | | | | | | 184 |

AG= Angular Gyrus
TP= temporal lobe
IFG= Inferior frontal gyrus
STG= superior temporal gyrus
MFG=middle fontal gyrus
postcg=postcentral gyrus
aMTG=anterior middle temporal gyrus
pMTG=posterior middle temporal gyrus

## A.3 GENERALIZATION OF SRM IN NEURAL SPACE AND THE CORRESPONDING ENCODING PERFORMANCE

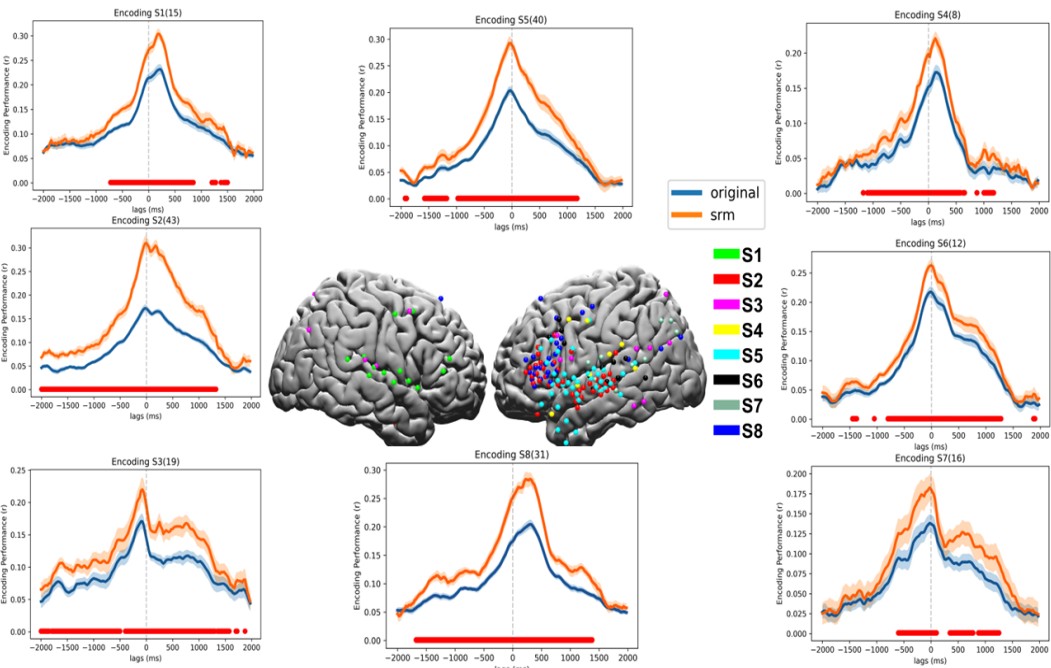

Figure 6: Encoding Performance in neural space of SRM-denoised data where SRM has been generalized to a new subject. The red horizontal line indicates the significant lags (p < 0.01, FDR corrected)

## A.4 Encoding Analysis for different layers of LLM

We extract contextual embeddings from all 48 layers of GPT2-XL and carried out our encoding analysis for both neural and shared responses at lags ranging from –2 s to +2 s relative to word onset. Figure 7 shows the final result. In both cases, we see that intermediate layers yield the highest prediction performance in human brain activity. This is consistent with a number of studies suggesting that the late-intermediate layers best match the human brain, and that there is no obvious mapping between different layers and different brain areas ( Schrimpf et al. (2021), Goldstein et al. (2023a), Caucheteux & King (2022), Kumar et al.). Furthermore, we see that shared space encoding is significantly higher than the original neural signal encoding.

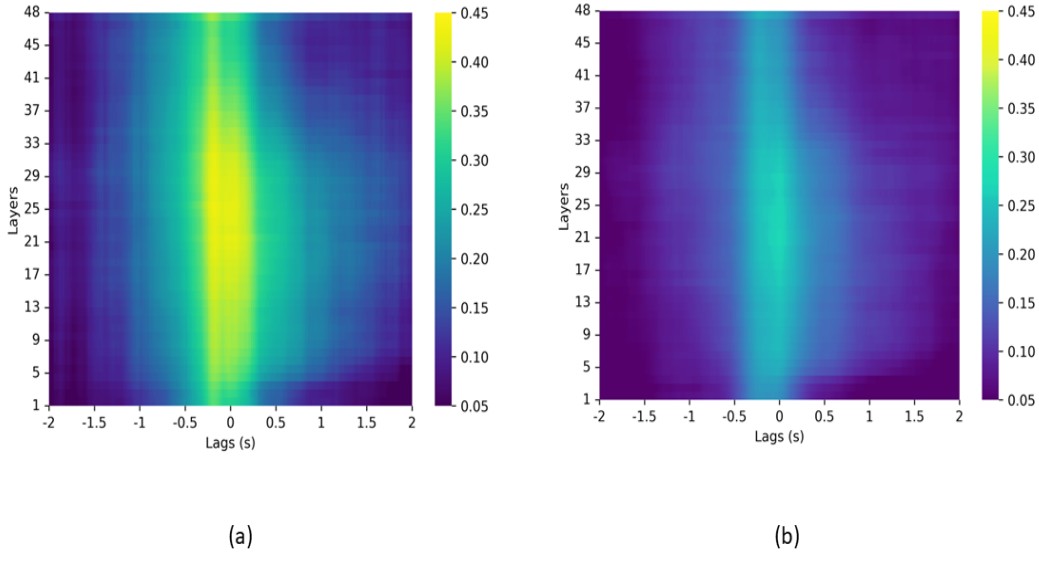

Figure 7: (a) Encoding analysis for shared space for all the layers of GPT-2-XL (b) Encoding analysis for original neural data for all the layers of GPT-2-XL

## A.5 ENCODING ANALYSIS FOR DIFFERENT SIZES OF LLM

We run our analysis for open source GPT models ranging from 125M to 20B parameters: neo-125M, large-774M, neo-1.3B, XL-1.5B, neo-2.7B, neo-20B. We do this analysis for all the subjects and for both neural and shared space encoding. Figure 8 shows the final result. In keeping with prior work (Antonello et al. (2023)), we observe a weak relationship where encoding models with larger parameters may yield increased encoding performance. If we plot the maximum correlation across lags with respect to the number of model parameters, we observe a log-linear relationship.

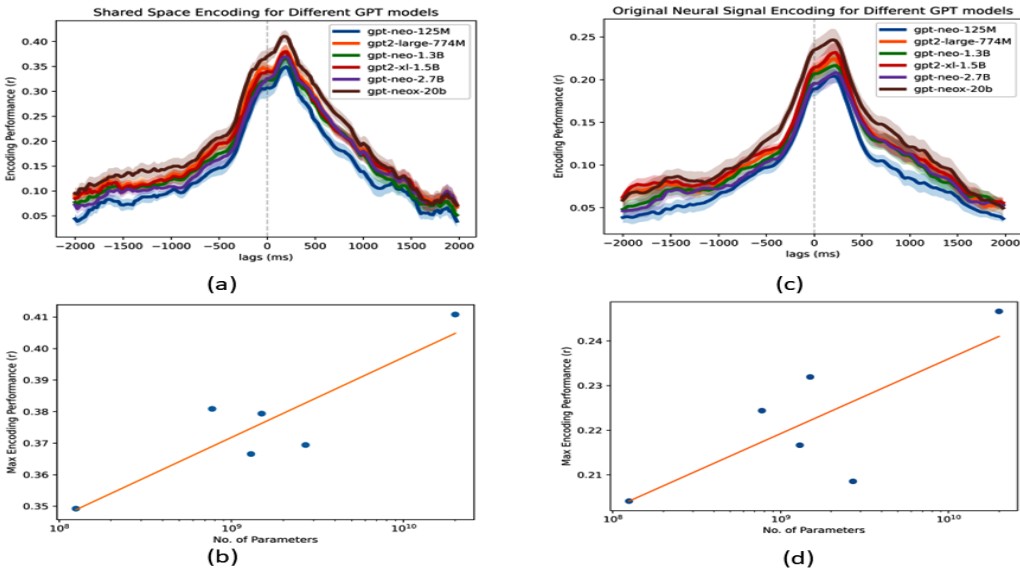

Figure 8: (a) Encoding analysis for shared space for different sizes of GPT model (b) log-linear scaling of encoding model performance for shared space with the number of parameters for different GPT models (c) Encoding analysis for original neural data for different sizes of GPT model (b) log-linear scaling of encoding model performance for original neural data with the number of parameters for different GPT models

## A.6 IMPACT OF SUBJECT SIZE FOR SRM

We design an experiment to see how SRM reconstruction from other subjects performs under various training subject sizes. We carry out the analysis explained in section 3.5 for each of the 8 subjects in the dataset, where the SRM training subject size is varied from 2 subjects (the minimal number) to 7 subjects (the full sample). We compute the correlation between the test subject's actual neural activity and the neural activity reconstructed from the other subjects. The subsets of training subjects are randomly resampled 5 times. We report the mean reconstruction correlation across subjects at each number of training subjects in Figure 9. We find that the reconstruction quality increased with larger sample sizes.

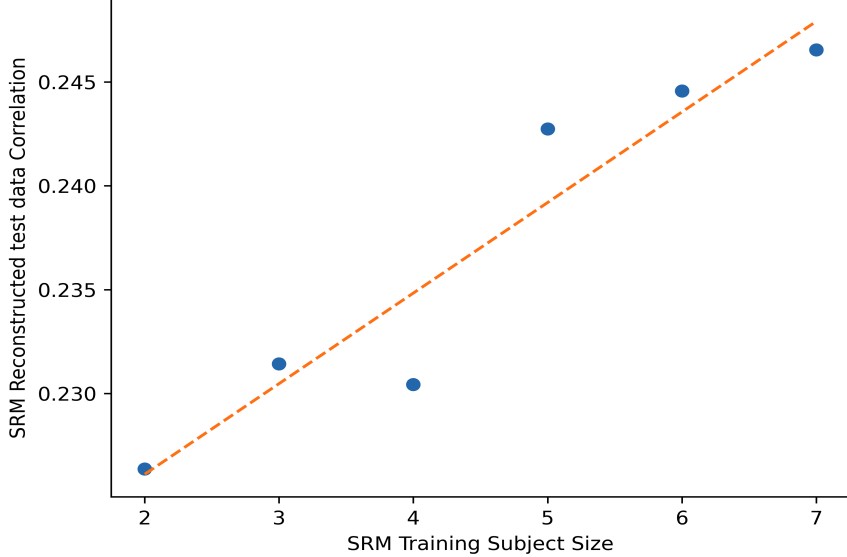

Figure 9: SRM reconstruction correlation for different SRM training subject size

### A.7 ENCODING ANALYSIS FOR DIFFERENT TYPES OF WORD EMBEDDINGS

We run the encoding analysis for two types of word embeddings- contextual (GPT2-XL) and static (GloVe) embeddings (Pennington et al. (2014)). We carry out the analyses for both shared space encoding and original neural signal encoding. Figure 10 shows the result and we can see that for both shared space and neural signal encoding, the contextual embeddings are giving a significantly higher encoding performance compared to the static embeddings. Furthermore, for both contextual and static embeddings, shared space encoding is significantly higher than that of the neural signal. This fits with the prior literature comparing static and contextual embeddings (Goldstein et al. (2022b)), and demonstrates that the improvement due to SRM is not specific to the encoding model based on contextual embeddings.

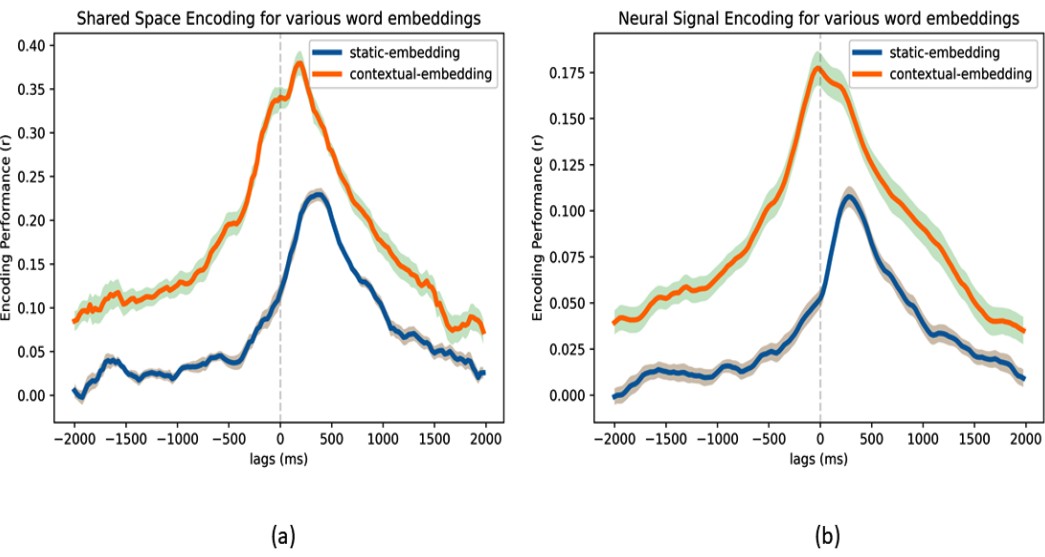

Figure 10: Encoding analysis for different types of word embeddings– (a) shared space encoding (b) original neural signal encoding

### A.8 DIFFERENTIATING BETWEEN GOOD OR BAD SUBJECTS

In section 3.5, we introduce an analysis to see how much information is shared across subjects. We reconstruct the neural signal for a left-out subject j on the left-out segment of the stimulus using the data of other subjects, and quantify how well this correlates with the actual neural signal for that test subject j and test stimulus. From the Table 2, we can see that SRM allows us to reconstruct left-out subject data at greater than 0.20 correlation (based on other subjects' data) for all test subjects, and their encoding analyses (Figure 1(b)) also confirm that they are giving good performances. Based on this experiment, we extend this analysis to see how much information is shared if the data is noisy —indicative of a bad subject. We design a simulation to measure how accurately we can reconstruct neural activity for a given test subject under various signal to noise ratio (SNR) conditions. We deliberately introduce Gaussian noise into the test subject j's signal at various noise power, and for each level of SNR we measure how well SRM can reconstruct the test data from other (unperturbed) subjects. The hypothesis is that as we add more noise, the SRM based reconstruction from other subjects should do worse, and hence the shared information will degrade. Given this metric for varying noise levels, for a new subject, the correlation calculated by SRM based reconstruction should give us an idea of the subject's data quality.

Figure 11 shows the neural signal for the first 500 words for an example subject under different SNR conditions. The correlation between the SRM-based reconstructed signal and the neural signal at different SNR for all the subjects is given in the Table 5. We see that as the noise power is increased, the SRM based reconstruction gets poorer, as expected. Hence, This experiment provides a heuristic for selecting a threshold correlation value for reconstruction in new subjects; if the correlation is low, this would be indicative of a "bad" subject.

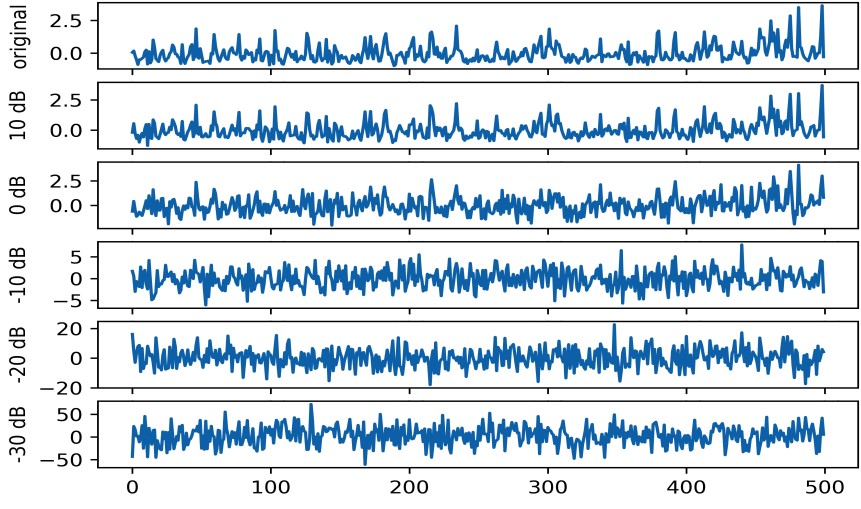

Figure 11: Neural signal for subject S8 at different SNR

Finally, we show that while SRM does improve encoding performance, it does not obscure which subjects are good or bad. Figure 12 demonstrates subjects with poor encoding performance prior to SRM also tend to have relatively poor (but improved) encoding performance after SRM. This improvement may relate to the number of electrodes in each subject, but this will require further research to fully understand.

Table 5: SRM Reconstrction Performance under different SNR

| SNR (dB) | Correlation with the original test data and the reconstructed data by SRM | | | | | | | |
|---|---|---|---|---|---|---|---|---|
| | **S1** | **S2** | **S3** | **S4** | **S5** | **S6** | **S7** | **S8** |
| original | 0.28 | 0.20 | 0.21 | 0.23 | 0.28 | 0.32 | 0.19 | 0.23 |
| 10 | 0.26 | 0.19 | 0.20 | 0.22 | 0.27 | 0.3 | 0.18 | 0.22 |
| 0 | 0.19 | 0.14 | 0.15 | 0.16 | 0.19 | 0.21 | 0.14 | 0.15 |
| -10 | 0.07 | 0.05 | 0.06 | 0.06 | 0.07 | 0.09 | 0.05 | 0.06 |
| -20 | 0.02 | 0.01 | 0.01 | 0.017 | 0.015 | 0.02 | 0.02 | 0.02 |
| -30 | 0.006 | 0.002 | -0.004 | -0.006 | -0.002 | 0.0015 | 0.003 | 0.0008 |

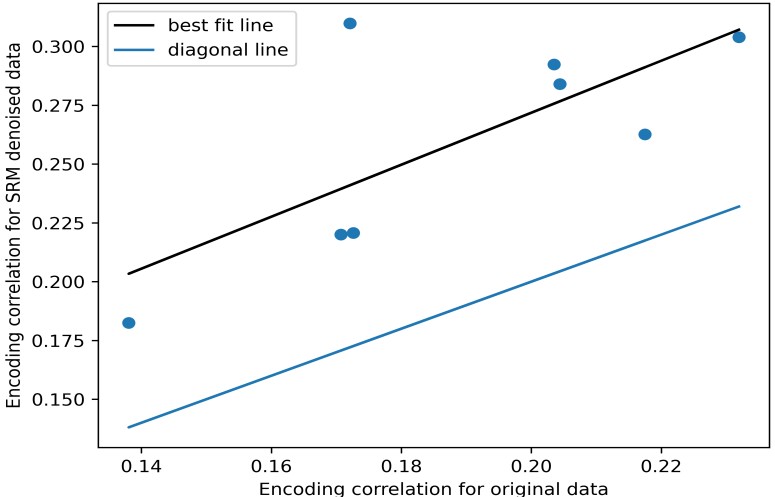

Figure 12: Subject wise comparison of encoding performance for original and SRM encoding

### A.9   ANALYSIS WITH A BEHAVIORAL EXPERIMENT AND DIFFERENT STIMULUS PROPERTIES

Based on the analysis section 3.2, we find that SRM-denoising improves encoding performance for all the subjects. However, we want to see whether the words that are more predictable across subjects, achieve higher improvements. For this purpose, we use the data of a behavioral experiment where for each word, we have a predictability score from 50 subjects. The underlying assumption is that the words which are more predictable, the brain responses are expected to be more shared (Nguyen et al. (2019)), and hence, SRM may incorporate a better improvement. In Fig. 13, we show the relative encoding performance improvement with SRM for all subjects for three word predictability criteria. We see that words with higher predictability show more significant improvement using SRM-denoised encoding.

Next, we are interested to see how different stimulus properties encode information in brain responses. We split the words into content and function words and for each, we run encoding analyses with both the original data and the SRM-denoised data. Figure 14 and 15 in appendix show the corresponding results. We can see that for all subjects, encoding with SRM-denoised data improves performance for both the word types. Also, we find that the encoding correlation for content words is higher than that for function words. These content words carry the meaning of the sentences and hence, this may be a reason for neural data to have stronger encoding performance.

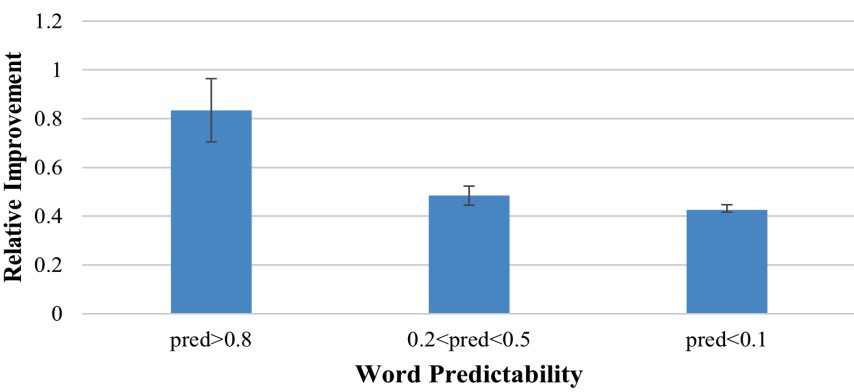

Figure 13: SRM-denoising based encoding for different word predictability criteria. From a behavioral experiment, we have a predictability score for each word. We carry out SRM-denoised encoding for three different word predictability criteria and the figure shows the relative performance improvement compared to original encoding.

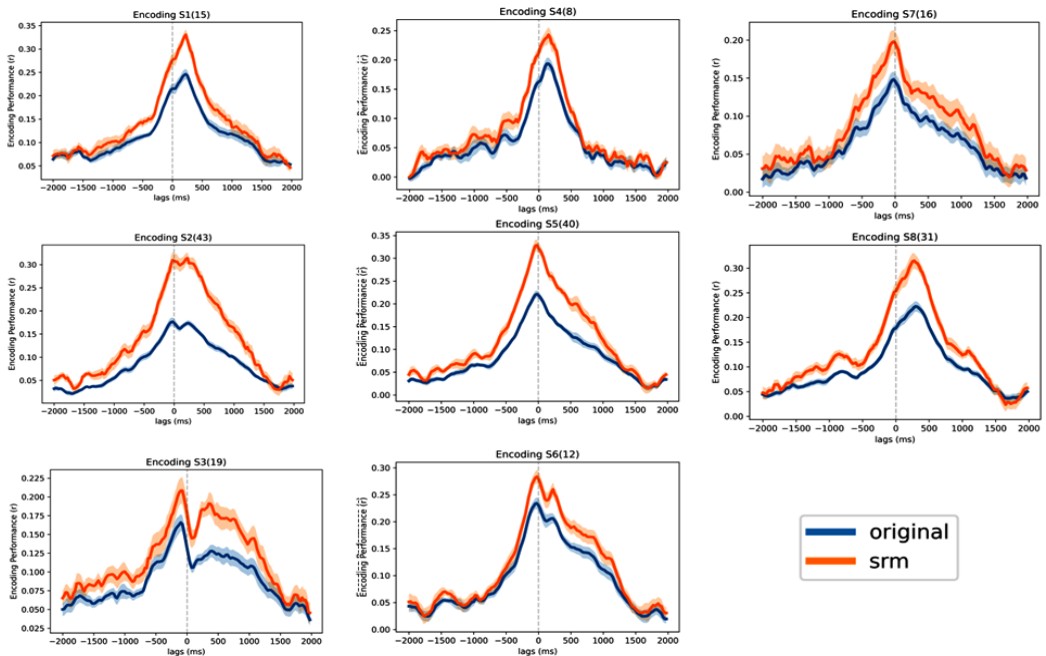

Figure 14: Encoding for content words

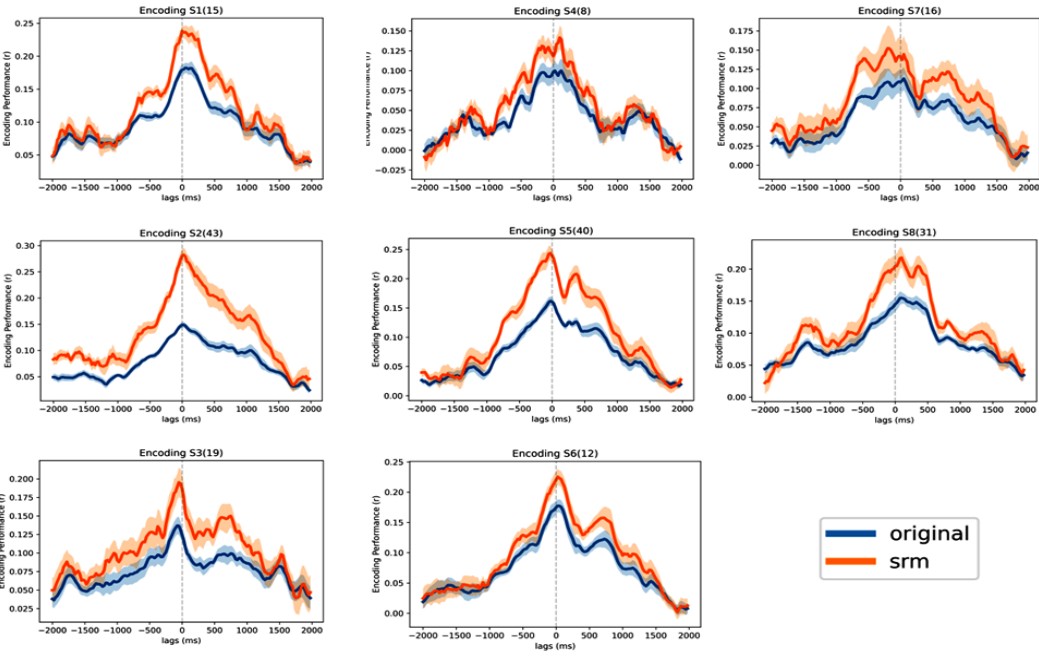

Figure 15: Encoding for function words

### A.10  USING RIDGE REGRESSION FOR ENCODING ANALYSIS

We use PCA to reduce the dimensionality of embeddings based on the precedent set by prior work (Goldstein et al. (2022b)), because absolute model performance is not the focus of this work, and to increase computational efficiency. We re-estimate encoding the encoding models at full dimensionality using ridge regression (which is commonly used in related work; e.g.Huth et al. (2016)). The regularization term in ridge regression will effectively reduce the dimensionality of the predictor matrix as well, but in a supervised fashion that should yield better overall performance. In keeping with this expectation, we observed marginally higher encoding performance using ridge regression at full dimensionality as compared to the original encoding analysis using ordinary least squares with the PCA-reduced embeddings (Fig. 16).

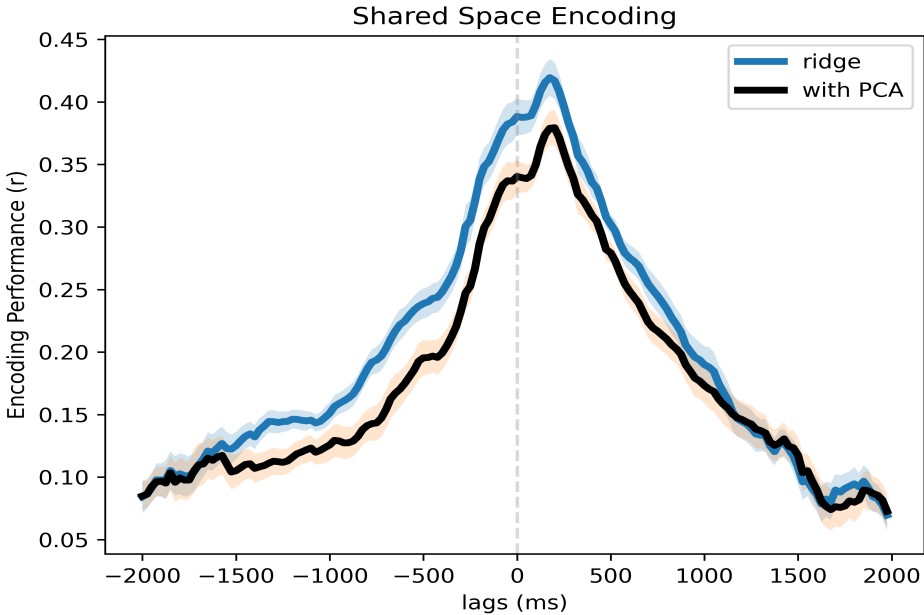

Figure 16: Comparison of Encoding Analysis using Ridge regression and PCA-reduced embeddings

