# OpenReview forum: "Aligning Brains into a Shared Space Improves Their Alignment to Large Language Model"
_ICLR.cc/2024/Conference — Submitted to ICLR 2024_

### Official Review · Reviewer_CbtZ · 2023-10-19

**Soundness:** 2 fair
**Presentation:** 2 fair
**Contribution:** 1 poor
**Rating:** 5
**Confidence:** 1

**Summary:**

In this work the researchers aim to understand human language processing by building encoding models predicting neural responses based on LLM contextual embeddings. Studying shared stimuli interpretations among subjects, the study investigates common information spaces using LLM-based encoding models and compares them with original brain responses. The research uses neural responses from eight subjects listening to a podcast, employing a Shared Response Model (SRM) to aggregate data and identify shared stimulus features. Results show significantly improved encoding performance using the SRM-calculated shared space compared to individual brain data. This shared space is also used to denoise individual responses, leading to substantial performance enhancements. The study demonstrates generalizability to new subjects and highlights the effectiveness of shared space in isolating stimulus-related features, particularly in brain areas specialized for language comprehension, offering deeper insights into human language processing.

**Strengths:**

The combination of utilizing Large Language Models (LLMs) and the Shared Response Model (SRM) to explore common information spaces in human brain processing seems to be a novel approach. The significant improvements in encoding performance and the ability to denoise individual responses using shared space, are noteworthy.

**Weaknesses:**

The paper lacks clarity regarding whether it introduces new methods or replicates Goldstein et al.'s (2022) approach. This ambiguity hampers a clear understanding of the paper's contributions.

The paper misses an opportunity to explore the impact of scale by not conducting the study on smaller auto-regressive models. With the vast parameter difference between GPT-2 XL and GPT-4, assessing the performance of smaller models would provide valuable insights into scalability and generalizability.

The impact of findings or what they entail in this field of study is not clearly described. For example other works have been carried out studying shared response for fMRI.

**Questions:**

Is the paper introducing any new methods, or is it simply replicating the approach outlined in Goldstein et al. (2022)? The methodology section lacks clarity, making it unclear whether the same setup as Goldstein et al. is being utilized. It is crucial to differentiate your work from previous research and provide appropriate citations for the foundational work you are building upon.

It would have been valuable to observe the study conducted on smaller auto-regressive models to assess the impact of scale (i.e. using large models). Btw, GPT-2 XL, with 1.5 billion parameters, in comparison to GPT-4's 1.76 trillion parameters, raises questions about whether it can still be classified as a Language Learning Model (LLM) today.

Why do you need to apply dimensionality reduction to the extracted representation? Does that not reduce the representation power of the model? There has been work showing that applying PCA to the repsentations achieve sub-optimal performance. Also some of the relevant knowledge may not be present at the final layer. Did you look at the representations from the earlier layers?

---

> ### Author Response · Authors · 2023-11-22
>
> **Weakness 3:** *The impact of findings or what they entail in this field of study is not clearly described. For example other works have been carried out studying shared response for fMRI.*
>
> In response to Weakness 3, we have tried to clarify the motivation and novelty of our research in the revised manuscript. The novelty of our paper is in applying the shared response model (SRM) to better aggregate electrode signals across brains, and showing how this benefits encoding model performance. Hyperalignment methods, including SRM, were developed in the context of fMRI (e.g. Haxby et al., 2020), where there is already a correspondence between features across subjects: all subjects are typically scanned using the same image acquisition parameters and thus have the same number of corresponding voxels. Most prior applications of hyperalignment were not evaluated in terms of encoding model performance, with some exceptions (e.g. Van Uden, Nastase et al., 2018; Nastase et al., 2020). Unlike prior results from fMRI, we show a large improvement in encoding performance with SRM.
>
> ECoG presents a more difficult correspondence problem than fMRI because each patient has a different number of electrodes in different locations (chosen for clinical reasons, not for research). In most ECoG research, including work by Goldstein et al., 2022, electrodes are simply combined across subjects to construct a “supersubject”. No effort is made to find a mapping from one subject to another—and, critically, no effort is made to ensure that encoding models actually generalize across subjects. How to best aggregate electrodes across subjects is a matter of ongoing research (e.g. Owen et al., 2020). To move beyond work by Goldstein et al., 2022, we use SRM to learn a reduced-dimension shared feature space across subjects that allows us to build encoding models that generalize from training subjects to left-out test subjects. Finally, in the revised manuscript, we include a method for quantifying how well the SRM can reconstruct a test subject’s data based on other subjects. This metric can serve as a “noise ceiling” for estimating how much reliable variance in the data is available for modeling.
>
> We have added a summary of this response in the Discussion section to better describe the impact of the findings. We also have re-written our abstract and introduction to reflect on this.
>
> *Reference:*
>
> Haxby, J. V., Guntupalli, J. S., Nastase, S. A., & Feilong, M. (2020). Hyperalignment: Modeling shared information encoded in idiosyncratic cortical topographies. elife, 9, e56601.
>
> Van Uden, C. E., Nastase, S. A., Connolly, A. C., Feilong, M., Hansen, I., Gobbini, M. I., & Haxby, J. V. (2018). Modeling semantic encoding in a common neural representational space. Frontiers in neuroscience, 12, 437.
>
> Nastase, S. A., Liu, Y. F., Hillman, H., Norman, K. A., & Hasson, U. (2020). Leveraging shared connectivity to aggregate heterogeneous datasets into a common response space. NeuroImage, 217, 116865.
>
> Goldstein, A., Zada, Z., Buchnik, E., Schain, M., Price, A., Aubrey, B., ... & Hasson, U. (2022). Shared computational principles for language processing in humans and deep language models. Nature neuroscience, 25(3), 369-380.
>
> Owen, L. L., Muntianu, T. A., Heusser, A. C., Daly, P. M., Scangos, K. W., & Manning, J. R. (2020). A Gaussian process model of human electrocorticographic data. Cerebral Cortex, 30(10), 5333-5345.

---

> ### Author Response · Authors · 2023-11-22
>
> **Question 1:** *Is the paper introducing any new methods, or is it simply replicating the approach outlined in Goldstein et al. (2022)? The methodology section lacks clarity, making it unclear whether the same setup as Goldstein et al. is being utilized. It is crucial to differentiate your work from previous research and provide appropriate citations for the foundational work you are building upon.*
>
> We appreciate this comment and have tried to clarify how our work differs from prior work. Goldstein et al. (2022) demonstrate that contextual embeddings derived from an autoregressive LLM can predict neural activity for the context-specific meaning of words in real-world contexts. As previously stated in the response of *Weakness 3*, electrodes are simply combined across subjects and all encoding analyses are performed separately per electrode (i.e. within each subject). In this work, we introduce a novel application of SRM for aggregating idiosyncratic electrode-wise signals to support estimating encoding models that generalize across subjects. We have adjusted the text of the revised manuscript to clarify how our methodology provides an advance over prior work by allowing us to build encoding models that predict brain activity in left-out subjects.
>
> We have re-written our abstract and introduction to reflect on this. We also provide a discussion on this in the Discussion section of the updated manuscript.
>
> *Reference:*
>
> Goldstein, A., Zada, Z., Buchnik, E., Schain, M., Price, A., Aubrey, B., ... & Hasson, U. (2022). Shared computational principles for language processing in humans and deep language models. Nature neuroscience, 25(3), 369-380.

---

> ### Author Response · Authors · 2023-11-22
>
> **Question 2:** *It would have been valuable to observe the study conducted on smaller auto-regressive models to assess the impact of scale (i.e. using large models). Btw, GPT-2 XL, with 1.5 billion parameters, in comparison to GPT-4's 1.76 trillion parameters, raises questions about whether it can still be classified as a Language Learning Model (LLM) today.*
>
> We thank the reviewer for suggesting that we explore the impact of model size. Unfortunately, it is not yet possible to extract word-by-word embeddings from GPT-3, GPT-3.5, and GPT-4. However, to address this comment we have replicated our analyses on several different open source GPT models ranging from 125M to 20B parameters: neo-125M, large-774M, neo-1.3B, XL-1.5B, neo-2.7B, neo-20B. In Figure 8 of Appendix A.5, we plot encoding performance both with and without SRM for models of increasing size. In keeping with prior work (Antonello et al., 2023), we observe a weak relationship where encoding models with larger parameters may yield increased encoding performance. If we plot the maximum correlation across lags with respect to the number of model parameters, we observe a log-linear relationship. Note that in this analysis, we still reduce the dimensionality of each model using PCA prior to computing encoding performance for the sake of model comparison and computational efficiency. This, however, may negatively impact the larger models more severely, and future work is needed to explore this issue more thoroughly. In the following comment, we demonstrate that retaining the original dimensionality and estimating encoding models using ridge regression marginally improves performance.
>
> In the revised manuscript, we discuss about this analysis in the Discussion section and provide the detailed analysis in the appendix section A.5.
>
> *Reference:*
>
> Antonello, R., Vaidya, A., & Huth, A. G. (2023). Scaling laws for language encoding models in fMRI. arXiv preprint arXiv:2305.11863.

---

> ### Author Response · Authors · 2023-11-22
>
> **Question 3** *Why do you need to apply dimensionality reduction to the extracted representation? Does that not reduce the representation power of the model? There has been work showing that applying PCA to the repsentations achieve sub-optimal performance.*
>
> For the original submission, we used PCA to reduce the dimensionality of embeddings based on the precedent set by prior work (e.g. Goldstein et al., 2022), because absolute model performance is not the focus of this work, and to increase computational efficiency. Note that Goldstein et al., 2022, used PCA specifically to match dimensionality when comparing embeddings from different language models (e.g. GPT-2 versus GloVe). The reviewer is completely correct that this unsupervised dimensionality reduction will tend to lower absolute model performance. To address the reviewer’s comment, we re-estimated the encoding models at full dimensionality using ridge regression (which is commonly used in related work; e.g. Huth et al., 2016). The regularization term in ridge regression will effectively reduce the dimensionality of the predictor matrix as well, but in a supervised fashion that should yield better overall performance. In keeping with this expectation, we observed marginally higher encoding performance using ridge regression at full dimensionality as compared to the original encoding analysis using ordinary least squares with the PCA-reduced embeddings (Fig. 16).
>
> *Reference:*
>
> Goldstein, A., Zada, Z., Buchnik, E., Schain, M., Price, A., Aubrey, B., ... & Hasson, U. (2022). Shared computational principles for language processing in humans and deep language models. Nature neuroscience, 25(3), 369-380.
>
> Huth, A. G., De Heer, W. A., Griffiths, T. L., Theunissen, F. E., & Gallant, J. L. (2016). Natural speech reveals the semantic maps that tile human cerebral cortex. Nature, 532(7600), 453-458.

---

> ### Author Response · Authors · 2023-11-22
>
> **Question 3** *Also some of the relevant knowledge may not be present at the final layer. Did you look at the representations from the earlier layers?*
>
> We appreciate the reviewer’s suggestion to examine different layers of the LLM. We extracted contextual embeddings from all 48 layers of GPT2-XL and carried out our encoding analysis for both neural and shared responses at lags ranging from –2 s to +2 s relative to word onset. Figure 7 in appendix section A.4 in the updated manuscript shows the final result. In both cases, we see that intermediate layers yield the highest prediction performance in human brain activity. This is consistent with a number of studies suggesting that the late-intermediate layers best match the human brain, and that there is no obvious mapping between different layers and different brain areas (Schrimpf et al., 2021; Caucheteux & King, 2022; Goldstein, Ham et al., 2022; Kumar, Sumers et al., 2022). Furthermore, we see that shared space encoding is significantly higher than the original neural signal encoding.
>
> In the revised manuscript, we discuss about this analysis in the Discussion section and provide the detailed analysis in the appendix section A.4.
>
> *Reference:*
>
> Schrimpf, M., Blank, I. A., Tuckute, G., Kauf, C., Hosseini, E. A., Kanwisher, N., ... & Fedorenko, E. (2021). The neural architecture of language: Integrative modeling converges on predictive processing. Proceedings of the National Academy of Sciences, 118(45), e2105646118.
>
> Caucheteux, C., & King, J. R. (2022). Brains and algorithms partially converge in natural language processing. Communications biology, 5(1), 134.
>
> Goldstein, A., Ham, E., Nastase, S. A., Zada, Z., Grinstein-Dabus, A., Aubrey, B., ... & Hasson, U. (2022). Correspondence between the layered structure of deep language models and temporal structure of natural language processing in the human brain. BioRxiv, 2022-07.
>
> Kumar, S., Sumers, T. R., Yamakoshi, T., Goldstein, A., Hasson, U., Norman, K. A., ... & Nastase, S. A. Shared functional specialization in transformer-based language models and the human brain. bioRxiv 2022.06.08.495348; doi: https://doi.org/10.1101/2022.06.08.495348

---

### Official Review · Reviewer_MBTm · 2023-10-31

**Soundness:** 3 good
**Presentation:** 3 good
**Contribution:** 3 good
**Rating:** 6
**Confidence:** 3

**Summary:**

This paper offers new insights into how our brains understand language. This work examines the neural responses of eight subjects as they listened to a podcast. Using a method called the Shared Response Model (SRM) on each subject's neural data, a common shared information space is learned across subjects. This shared information space, along with the contextual embeddings for each word in the podcast, is then used to build an encoding model. Experimental results indicate that the shared space derived from the SRM provides better encoding performance than the original brain responses.

**Strengths:**

1. Clear description of background knowledge and motivations needed to understand the study.
2. Clear exposition of the proposed model.
3. The paper demonstrates that aligning brain responses across subjects into a shared space significantly enhances the encoding performance.
4. Experimental demonstrates that the SRM model is generalizable to new subjects suggesting the model's robustness and adaptability.

**Weaknesses:**

1. The paper lacks implementation details under the Shared Response Model (SRM). How were S and W jointly estimated? What method was used to estimate W? Was it solved under the orthogonal Procrustes problem?
2. Could the experimental results be compared against other baselines?

**Questions:**

1. So what happens in the case where you don't have initial parallel data between the two spaces under the Shared Response Model (SRM) could the linear transformation (W) be learned in an unsupervised through optimal transport?
2. How does the Shared Response Model (SRM)  scale to a larger subject size?

---

> ### Author Response · Authors · 2023-11-21
>
> **Weakness 1:** *The paper lacks implementation details under the Shared Response Model (SRM). How were S and W jointly estimated? What method was used to estimate W? Was it solved under the orthogonal Procrustes problem?*
>
> In the original implementation of SRM (Chen et al., 2015), the problem is extended to a probabilistic setting. At time t, the equation stands
> $x_t=Ws_t+\mu+ \epsilon$, where $x_t \in \mathbb{R}^{mv}$, $s_t \in \mathbb{R}^{k}$ [m=subject size, v=voxel number, k= SRM hyperparameter]
>
> The model stands as
> $s_t \sim \cal{N} (0,\Sigma_s)$
>
> $x_{it}|s_t \sim \cal{N} (W_{i}s_{t}+\mu_i,\rho^2\cal{I})$ [ $x_{it}$ is the response at time t for subject $i$,  $\rho^2$ is a subject independent hyperparameter]
>
> $W_{i}^TW_i=I_k$ [$W_i \in \mathbb{R}^{v \times k}$]
>
> The S and W parameters of the SRM model are jointly estimated using a constrained EM algorithm. The complete implementation details can be found in Section 3 of Chen et al., 2015.
>
>
> *References:*
>
> Chen, P. H. C., Chen, J., Yeshurun, Y., Hasson, U., Haxby, J., & Ramadge, P. J. (2015). A reduced-dimension fMRI shared response model. Advances in Neural Information Processing Systems, 28.

---

> ### Comment · Reviewer_MBTm · 2023-11-22
> **Comments after rebuttal**
>
> Thank you for your response. I am revising my overall recommendation to 6.

---

> > ### Author Response · Authors · 2023-11-22
> >
> > Thanks for your score increase. Now we have included response to all the questions and have uploaded an updated manuscript.

---

> ### Author Response · Authors · 2023-11-22
>
> **Weakness 2:** *Could the experimental results be compared against other baselines?*
>
> We appreciate the reviewer highlighting this weakness. To address this concern, and concerns outlined by the other reviewers, we have included comparisons to several possible baselines in the revised manuscript: (1) original comparison between SRM and PCA baseline; (2) comparison between SRM and a simpler electrode-wise noise ceiling; (3) comparison between original contextual embeddings and static, non-contextual GloVe embeddings; as well as (4) comparisons across layers and model sizes. We outline each of these comparisons below.
>
> **(1) original comparison between SRM and PCA baseline:** First, we wanted to better articulate how our original principal component analyses (PCA) baseline serves as a control. Neural activity contains redundancies across electrodes, and the number and positioning of electrodes varies across subjects. The shared response model (SRM) reduces the dimensionality of this signal into a number of roughly orthogonal components (in this manuscript, we set the SRM hyperparameter k = 5 components). Each of these shared features learned by the SRM is a linear combination of electrode signals. How best to compare these 5 orthogonal SRM components to the original multi-electrode signals? Based on the prior literature (Chen et al., 2015; Nastase et al., 2020), we use PCA as a baseline for comparison. PCA comprises the same dimensionality reduction with the same orthogonality constraint, but with one important difference: while SRM learns a rotation to map each subject into the shared space, PCA does not. We use PCA as a baseline because it mimics several characteristics of SRM, without the critical function alignment step. We find that SRM yields markedly higher encoding performance than PCA (Figure 1(a)).
>
> **(2) comparison between SRM and a simpler electrode-wise noise ceiling:** In response to other reviewers, we have now introduced a new analysis where we quantify how well each test subject’s neural activity can be reconstructed from other subjects (section 3.5). This reconstruction quality serves as a between-subject “noise ceiling” by resolving idiosyncrasies in feature spaces across subjects. We evaluated this reconstruction metric against a simpler estimate of the noise ceiling based on intersubject correlation proposed in prior work (Nastase et al., 2019). For each electrode in the test subject, we obtain the maximum correlation among all other electrodes across all other subjects, then compute the mean of these maximum correlations across electrodes. We repeat this process for all the subjects and report the results in Tables 2 and 3 in the main text of the revised manuscript. We see that the SRM reconstruction across subjects yields considerably higher correlations than naive noise ceiling estimates based electrode-wise intersubject correlations.
>
> **(3) comparison between original contextual embeddings and static, non-contextual GloVe embeddings:** In the revised manuscript, we now also compare the contextual word embeddings derived from GPT2-XL to a baseline of static, non-contextual embeddings derived from GloVe (Pennington et al., 2014; appendix section A.7). We qualitatively compare the encoding performance for both types of embeddings both with SRM and for the original neural signals (Fig. 10). In keeping with prior work (e.g. Goldstein et al., 2022), we find that the contextual embeddings outperform the static embeddings. Importantly, we also see that the SRM improves encoding performance for both types of embeddings.
>
> **(4) comparisons across layers and model sizes:** Finally, we include two more comparisons based on the comments of other reviewers. First, we compare encoding performance across layers of GPT2-XL (Fig. 7) and find that intermediate layers provide the best performance (appendix section A.4). Importantly, we also find that SRM improves encoding performance across all the layers. In the revised manuscript, we also compare encoding performance for several different sizes of models (Figure 8, appendix section A.5). We observed a weak relationship suggesting that encoding performance may improve for larger models (Antonello et al., 2023); importantly, however, we see that SRM improves performance across all models.

---

> ### Author Response · Authors · 2023-11-22
>
> *References of answer of Weakness 2:*
>
> Chen, P. H. C., Chen, J., Yeshurun, Y., Hasson, U., Haxby, J., & Ramadge, P. J. (2015). A reduced-dimension fMRI shared response model. Advances in Neural Information Processing Systems, 28.
>
> Nastase, S. A., Liu, Y. F., Hillman, H., Norman, K. A., & Hasson, U. (2020). Leveraging shared connectivity to aggregate heterogeneous datasets into a common response space. NeuroImage, 217, 116865.
>
> Nastase, S. A., Gazzola, V., Hasson, U., & Keysers, C. (2019). Measuring shared responses across subjects using intersubject correlation. Social Cognitive and Affective Neuroscience, 14(6), 667-685.
>
> Pennington, J., Socher, R., & Manning, C. D. (2014, October). Glove: Global vectors for word representation. In Proceedings of the 2014 conference on empirical methods in natural language processing (EMNLP) (pp. 1532-1543).
>
> Goldstein, A., Zada, Z., Buchnik, E., Schain, M., Price, A., Aubrey, B., ... & Hasson, U. (2022). Shared computational principles for language processing in humans and deep language models. Nature neuroscience, 25(3), 369-380.
>
> Antonello, R., Vaidya, A., & Huth, A. G. (2023). Scaling laws for language encoding models in fMRI. arXiv preprint arXiv:2305.11863.

---

> ### Author Response · Authors · 2023-11-22
>
> **Question 1:** *So what happens in the case where you don't have initial parallel data between the two spaces under the Shared Response Model (SRM) could the linear transformation (W) be learned in an unsupervised through optimal transport?*
>
> The SRM learns a linear transformation from the idiosyncratic feature spaces (i.e. electrodes) of each subject to a shared feature space in an unsupervised fashion (i.e. without respect to a target variable;  Yousefnezhad et al., 2020). Finding this mapping between features spaces, however, pivots on a shared axis determined by the number of samples corresponding to the same shared stimulus (e.g. an audiovisual movie, a spoken story). Prior research has in fact developed an optimal transport method for hyperalignment (Bazeille et al., 2019), but to our knowledge this algorithm has only been used with a matching set of samples (i.e. the same stimulus) across subjects.
>
> Recent work in fMRI, however, has introduced a hyperalignment method that does not require a shared number of samples for the same stimulus (Guntupalli et al., 2018; Nastase et al., 2020; Busch et al., 2021). In this work, rather than using the time-locked response time series to drive alignment, the authors first estimate “functional connectivity”—i.e. correlations between activity time series—between the features of interest and a fixed set of connectivity targets shared across subjects (e.g. searchlight or parcel time series across the whole brain). This fixed set of connectivity targets can be derived within each subject based on time series that are of different lengths or corresponding to different stimuli/tasks from subject to subject. These shared connectivity targets are then used to drive functional alignment of the features of interest. However, these connectivity-based hyperalignment algorithms rely on a shared set of connectivity targets that are easily obtained in fMRI—but which are typically not available in ECoG. Further work is needed to determine whether ECoG data can be aligned without a shared stimulus.
>
> *Reference:*
>
> Yousefnezhad, M., Selvitella, A., Han, L., & Zhang, D. (2020). Supervised hyperalignment for multisubject fmri data alignment. IEEE Transactions on Cognitive and Developmental Systems, 13(3), 475-490.
>
> Bazeille, T., Richard, H., Janati, H., & Thirion, B. (2019). Local optimal transport for functional brain template estimation. In Information Processing in Medical Imaging: 26th International Conference, IPMI 2019, Hong Kong, China, June 2–7, 2019, Proceedings 26 (pp. 237-248). Springer International Publishing.
>
> Guntupalli, J. S., Feilong, M., & Haxby, J. V. (2018). A computational model of shared fine-scale structure in the human connectome. PLoS computational biology, 14(4), e1006120.
>
> Nastase, S. A., Liu, Y. F., Hillman, H., Norman, K. A., & Hasson, U. (2020). Leveraging shared connectivity to aggregate heterogeneous datasets into a common response space. NeuroImage, 217, 116865.
>
> Busch, E. L., Slipski, L., Feilong, M., Guntupalli, J. S., di Oleggio Castello, M. V., Huckins, J. F., ... & Haxby, J. V. (2021). Hybrid hyperalignment: A single high-dimensional model of shared information embedded in cortical patterns of response and functional connectivity. NeuroImage, 233, 117975.

---

> ### Author Response · Authors · 2023-11-22
>
> **Question 2:** *How does the Shared Response Model (SRM) scale to a larger subject size?*
>
> In the literature, SRM has been used with a varying number of subjects depending on the dataset size. For example, in the original SRM paper (Chen et al., 2015), the authors used four datasets of varying sample size: 16,10,18 and 40. More recent work has extended SRM to samples comprising over 200 subjects (Nastase et al., 2020). This combined with benchmarking work suggesting that SRM requires relatively low computation time compared to competing methods (Bazeille, Dupre et al., 2021) suggests that the algorithm will scale to larger samples.
>
> In our study, due to the inherent limitations of ECoG acquisition, we have only 8 subjects. To address the reviewer’s comment, we devised an experiment to investigate how different numbers of subjects will impact the quality of the shared response model. We re-estimated the SRM for each of the 8 test subjects in the dataset, where the SRM was trained on a number of subjects ranging from 2 subjects (the minimal number) to 7 subjects (the full sample). We evaluated the quality of the SRM using the new reconstruction analysis described in section XX of the revised manuscript: we computed the correlation between the test subject’s actual neural activity and the neural activity reconstructed from the other subjects. The subsets of training subjects were randomly resampled 5 times. We report the mean reconstruction correlation across subjects at each number of training subjects in Figure 9 in Appendix A.6. We found that the reconstruction quality increased with larger sample sizes (although we may expect this to asymptote at even larger samples unavailable here).
>
> *References:*
>
> Chen, P. H. C., Chen, J., Yeshurun, Y., Hasson, U., Haxby, J., & Ramadge, P. J. (2015). A reduced-dimension fMRI shared response model. Advances in Neural Information Processing Systems, 28.
>
> Nastase, S. A., Liu, Y. F., Hillman, H., Norman, K. A., & Hasson, U. (2020). Leveraging shared connectivity to aggregate heterogeneous datasets into a common response space. NeuroImage, 217, 116865.
>
> Bazeille, T., Dupre, E., Richard, H., Poline, J. B., & Thirion, B. (2021). An empirical evaluation of functional alignment using inter-subject decoding. NeuroImage, 245, 118683.

---

### Official Review · Reviewer_6UCh · 2023-11-02

**Soundness:** 2 fair
**Presentation:** 2 fair
**Contribution:** 2 fair
**Rating:** 3
**Confidence:** 4

**Summary:**

This study aims to investigate whether the shared information space of subjects' brain data can be more accurately predicted than the original brain data using contextual embeddings derived from large language models like GPT-2. The primary contributions can be summarized as follows: a language encoding model was trained to predict original neural story responses and also predict shared neural responses across subjects using a shared response model derived from story representations. This resulted in a higher predictive accuracy for shared responses compared to original responses.
Subsequently, the language encoding model was employed to attempt the prediction of denoised individual brain responses by projecting them back into the neural space. The accuracy of predictions improved for denoised responses compared to the original responses. A detailed analysis of language ROI (Region of Interest) revealed that the superior temporal gyrus (STG) and inferior frontal gyrus (IFG) showed improvements over other regions.

**Strengths:**

1.	The idea of shared response model is a well-known technique and authors demonstrate that aligning brains into a shared space significantly improves the encoding performance.
2.	Additionally, the authors' examination of their encoding model's performance on ECoG recordings is intriguing, considering that prior studies predominantly concentrated on fMRI.

**Weaknesses:**

1.	The novelty in the paper is somewhat limited as it primarily delves into a comparison between shared responses, original responses, and denoised responses. Notably, the shared response model itself isn't a new concept.
2.	The implications of this paper are unclear to me. Given the authors' emphasis on shared responses rather than original neural responses, the role of contextual word representations remains ambiguous. If the authors intended to present findings related to Large Language Models (LLMs), they could have explored a comparison of different word embeddings such as GloVe or Word2Vec alongside word representations from LLMs to assess their performance. However, the authors solely utilized word representations from LLMs and concluded that LLMs performed better.
3.	Similarly, authors can explain what the percentage of shared information across subjects across regions is. Also, the comparison of estimated noise ceiling vs. shared responses is more interesting.
4.	The authors concentrate solely on contextual word representations from LLMs, but it's worth noting that there exists a rich language-hierarchy across layers, and layer-wise representations contain a wealth of information (such as POS tags, NER, and dependency tags) that may also be pertinent to neural brain responses. These findings provide substantial evidence across various types of information and shared responses among individual subjects.
5.	the clarity can be improved:
6.	Figure 6 result is difficult to parse because it contain a lot of information. Also, check the typo in Figure 6: Relaive -> Relative, Predictibility -> Predictability

**Questions:**

1. What the percentage of shared information across subjects across regions is? Is it possible to differentiate good vs. bad subjects?
2. The comparison of estimated noise ceiling vs. shared responses is more interesting.

---

> ### Author Response · Authors · 2023-11-22
>
> **Weakness 1:** *The novelty in the paper is somewhat limited as it primarily delves into a comparison between shared responses, original responses, and denoised responses. Notably, the shared response model itself isn't a new concept.*
>
> The novelty of our paper is in applying the shared response model (SRM) to better aggregate electrode signals across brains, and showing how this benefits encoding model performance. Hyperalignment methods, including SRM, were developed in the context of fMRI (e.g. Haxby et al., 2020), where there is already a correspondence between features across subjects: all subjects are typically scanned using the same image acquisition parameters and thus have the same number of corresponding voxels. Most prior applications of hyperalignment were not evaluated in terms of encoding model performance (with some exceptions; e.g. Van Uden, Nastase et al., 2018). ECoG presents a more difficult correspondence problem because each patient has a different number of electrodes in different locations (chosen for clinical reasons, not for research); how to best aggregate electrodes across subjects is a matter of ongoing research (e.g. Owen et al., 2020). Ultimately, these methods will allow us to leverage data from existing subjects to train encoding models that generalize to a new patient. We have tried to clarify the motivation and novelty in the revised text in the discussion section:
>
> *"Unlike fMRI, where there is already voxelwise feature correspondence, ECoG presents a more difficult correspondence problem because each subject has a different number of electrodes in different locations; how to best aggregate electrodes across patients is a matter of ongoing research (Owen et al., 2020)."*
>
> We also have re-written our abstract and introduction to reflect on this.
>
> *Reference:*
>
> Haxby, J. V., Guntupalli, J. S., Nastase, S. A., & Feilong, M. (2020). Hyperalignment: Modeling shared information encoded in idiosyncratic cortical topographies. elife, 9, e56601.
>
> Van Uden, C. E., Nastase, S. A., Connolly, A. C., Feilong, M., Hansen, I., Gobbini, M. I., & Haxby, J. V. (2018). Modeling semantic encoding in a common neural representational space. Frontiers in neuroscience, 12, 437.
>
> Owen, L. L., Muntianu, T. A., Heusser, A. C., Daly, P. M., Scangos, K. W., & Manning, J. R. (2020). A Gaussian process model of human electrocorticographic data. Cerebral Cortex, 30(10), 5333-5345.

---

> ### Author Response · Authors · 2023-11-22
>
> **Weakness 2:** *The implications of this paper are unclear to me. Given the authors' emphasis on shared responses rather than original neural responses, the role of contextual word representations remains ambiguous. If the authors intended to present findings related to Large Language Models (LLMs), they could have explored a comparison of different word embeddings such as GloVe or Word2Vec alongside word representations from LLMs to assess their performance. However, the authors solely utilized word representations from LLMs and concluded that LLMs performed better.*
>
> Our initial goal in using large language model (LLM) embeddings for the encoding analysis was to showcase how the SRM can improve encoding performance. However, we agree that the encoding results may be difficult to interpret without comparison to some baseline or other models. In the revised manuscript, we expanded the encoding analysis to also include static, non-contextual word embeddings from GloVe (Pennington et al., 2014). We carry out the encoding analyses for both the shared space encoding and original neural signal. Figure 10 demonstrates that for both shared space and neural signal encoding, the contextual embeddings yield significantly higher encoding performance compared to the static embeddings. Furthermore, for both contextual and static embeddings shared space encoding is significantly higher than that of the neural signal. This fits with the prior literature comparing static and contextual embeddings (e.g. Goldstein et al., 2022), and demonstrates that the improvement due to SRM is not specific to the encoding model based on contextual embeddings. We have included this new analysis in the appendix section A.7 of the revised manuscript.
>
> *Reference:*
>
> Pennington, J., Socher, R., & Manning, C. D. (2014, October). Glove: Global vectors for word representation. In Proceedings of the 2014 conference on empirical methods in natural language processing (EMNLP) (pp. 1532-1543).
>
> Goldstein, A., Zada, Z., Buchnik, E., Schain, M., Price, A., Aubrey, B., ... & Hasson, U. (2022). Shared computational principles for language processing in humans and deep language models. Nature neuroscience, 25(3), 369-380.

---

> ### Author Response · Authors · 2023-11-22
>
> **Weakness 3** *Similarly, authors can explain what the percentage of shared information across subjects across regions is. Also, the comparison of estimated noise ceiling vs. shared responses is more interesting.*
>
> This comment was very helpful, and we have developed a new analysis (Section 3.5 of the revised manuscript) to estimate how accurately we can reconstruct one subject’s neural signals based only on other subjects’ neural activity using SRM. There are many ways to estimate a “noise ceiling”—an estimate of the reliability of the data that provides an upper limit for how much variance a model might be able to capture—that fall into two general categories. The first method estimates reliability by resampling neural activity in the same subject based on repeated presentations of the same stimulus (e.g. Hsu et al., 2004; de Heer et al., 2017). This method requires data acquired during multiple repetitions of the same test stimulus; this is a weakness in the context of naturalistic paradigms given that repeated presentations of the same stimulus lose their novelty and are processed differently (e.g. subsequent presentations activate memory processes). The second method estimates reliability by using other subjects’ data for the same stimulus as a surrogate model (e.g. Nili et al., 2014; Nastase et al., 2019); this method retains the novelty of a naturalistic stimulus, but has the weakness of poor alignment or functional correspondence across individual brains. SRM in fact provides a way to estimating a higher, more robust noise ceiling using the second method because it improves functional correspondence across individuals.
>
> In order to establish a measure of shared information across subjects (without reference to any particular encoding model), we designed an experiment where we reconstruct the neural signal for a left-out subject j on the left-out segment of the stimulus using the data of other subjects, and quantify how well this correlates with the actual neural signal for that test subject j and test stimulus. Higher correlation indicates more shared information across subjects. For details please refer to the section 3.5 of the revised manuscript.
>
> We repeat this process for each subject for all the test folds at the word onset. Table 2 summarizes the result. From the Table we see that SRM based reconstruction from other subjects achieves a 0.25 correlation on average. That is, we can reconstruct one subject’s neural activity from other subjects’ neural activity at a 0.25 correlation; this is one formulation of a noise ceiling.
>
> How can we compare this SRM-based noise ceiling to a simpler noise ceiling used in prior work? We do not have multiple presentations of the same test stimulus from the same subject, and thus must rely on other subjects to estimate the noise ceiling. Furthermore, we do not have electrode correspondence across ECoG subjects (unlike voxels in fMRI). To circumvent these challenges, for a given test subject j, we obtain the electrode-wise signals for a set of test words and find the maximum correlation with all other electrodes across all other subjects. We compute the mean of these maximum intersubject correlations across electrodes and across subjects, and report the results in Table 2.  From the table, we see that SRM yields a higher, more robust noise ceiling across subjects than the simple electrode-wise method adopted from prior work. We believe this is an important analysis and have added it to the main text (section 3.5) of the revised manuscript.
>
> Based on the reviewer’s comment, we next addressed whether the amount of shared information across subjects differs across brain regions. We re-ran the above reconstruction analysis separately for two different brain regions: superior temporal gyrus (STG) and inferior temporal gyrus (IFG). For this analyses, we only considered subjects with five or more electrodes in the region of interest. Table 3 indicates better reconstruction across subjects in STG than in IFG, but both benefit from SRM at a similar magnitude.
>
> *Reference:*
>
> Hsu, A., Borst, A., & Theunissen, F. E. (2004). Quantifying variability in neural responses and its application for the validation of model predictions. Network: Computation in Neural Systems, 15(2), 91-109.
>
> de Heer, W. A., Huth, A. G., Griffiths, T. L., Gallant, J. L., & Theunissen, F. E. (2017). The hierarchical cortical organization of human speech processing. Journal of Neuroscience, 37(27), 6539-6557.
>
> Nili, H., Wingfield, C., Walther, A., Su, L., Marslen-Wilson, W., & Kriegeskorte, N. (2014). A toolbox for representational similarity analysis. PLoS computational biology, 10(4), e1003553.
>
> Nastase, S. A., Gazzola, V., Hasson, U., & Keysers, C. (2019). Measuring shared responses across subjects using intersubject correlation. Social Cognitive and Affective Neuroscience, 14(6), 667-685.

---

> ### Author Response · Authors · 2023-11-22
>
> **Weakness 4:** *The authors concentrate solely on contextual word representations from LLMs, but it's worth noting that there exists a rich language-hierarchy across layers, and layer-wise representations contain a wealth of information (such as POS tags, NER, and dependency tags) that may also be pertinent to neural brain responses. These findings provide substantial evidence across various types of information and shared responses among individual subjects.*
>
> We thank the reviewer for encouraging us to explore different layers of the LLMs. We extracted contextual embeddings from all 48 layers of GPT2-XL and carried out our encoding analysis for both neural and shared responses at lags ranging from –2 s to +2 s relative to word onset. Figure 7 in the updated manuscript shows the final result. In both cases, we see that intermediate layers yield the highest prediction performance in human brain activity. This is consistent with a number of studies suggesting that the late-intermediate layers best match the human brain, and that there is no obvious mapping between different layers and different brain areas (Schrimpf et al., 2021; Caucheteux & King, 2022; Goldstein, Ham et al., 2022; Kumar, Sumers et al., 2022). Furthermore, we see that shared space encoding is significantly higher than the original neural signal encoding.
>
> In the revised manuscript, we discuss about this analysis in the Discussion section and provide the detailed analysis in the appendix section A.4.
>
> *Reference:*
>
> Schrimpf, M., Blank, I. A., Tuckute, G., Kauf, C., Hosseini, E. A., Kanwisher, N., ... & Fedorenko, E. (2021). The neural architecture of language: Integrative modeling converges on predictive processing. Proceedings of the National Academy of Sciences, 118(45), e2105646118.
>
> Caucheteux, C., & King, J. R. (2022). Brains and algorithms partially converge in natural language processing. Communications biology, 5(1), 134.
>
> Goldstein, A., Ham, E., Nastase, S. A., Zada, Z., Grinstein-Dabus, A., Aubrey, B., ... & Hasson, U. (2022). Correspondence between the layered structure of deep language models and temporal structure of natural language processing in the human brain. BioRxiv, 2022-07.
>
> Kumar, S., Sumers, T. R., Yamakoshi, T., Goldstein, A., Hasson, U., Norman, K. A., ... & Nastase, S. A. Shared functional specialization in transformer-based language models and the human brain. bioRxiv 2022.06.08.495348; doi: https://doi.org/10.1101/2022.06.08.495348

---

> ### Author Response · Authors · 2023-11-22
>
> **Weakness 5:** *the clarity can be improved*
>
> We have revised the text throughout the manuscript in hopes of improving the clarity.
>
>
> **Weakness 6:** *Figure 6 result is difficult to parse because it contain a lot of information. Also, check the typo in Figure 6: Relaive -> Relative, Predictibility -> Predictability*
>
> Thanks! We have fixed the typos. Regarding Figure 6, we agree that this figure was unclear (and not particularly important to our core findings). We have clarified the analysis and have moved the figure out of the main text and into the appendix section A.9 (Figure 13).

---

> ### Author Response · Authors · 2023-11-22
>
> **Question 1:** *What the percentage of shared information across subjects across regions is?*
>
> We describe in detail in response to this query in *Weakness 3*
>
> **Question 1:** *Is it possible to differentiate good vs. bad subjects?*
>
> As we described in the response to Weakness 3 above, we have introduced a new analysis to quantify how much information is shared across subjects. We reconstruct the neural signal for a left-out subject j on the left-out segment of the stimulus using the data of other subjects, and quantify how well this correlates with the actual neural signal for that test subject j and test stimulus. From Table 2, we can see that SRM allows us to reconstruct left-out subject data at greater than 0.20 correlation (based on other subjects’ data) for all test subjects.
>
> Based on this experiment, we extend this analysis to see how much information is shared if the data are noisy—indicative of a bad subject. We designed a simulation to measure how accurately we can reconstruct neural activity for a given test subject under various signal to noise ratio (SNR) conditions. We deliberately introduce Gaussian noise into the test subject j’s signal at various noise power, and for each level of SNR we measure how well SRM can reconstruct the test data from other (unperturbed) subjects. The hypothesis is that as we add more noise, the SRM based reconstruction from other subjects should do worse, and hence the shared information will degrade. Given this metric for varying noise levels, for a new subject, the correlation calculated by SRM based reconstruction should give us an idea of the subject’s data quality.
>
> Figure 11 in appendix section A.8 shows the neural signal for the first 500 words for an example subject under different SNR conditions. The correlation between the SRM based reconstructed signal and the neural signal at different SNR for all the subjects is shown in the Table 5. We see that as noise increases, the SRM based reconstruction gets poorer, as expected. This experiment provides a heuristic for selecting a threshold correlation value for reconstruction in new subjects; if the correlation is low, this would be indicative of a “bad” subject.
>
> Finally, we show that while SRM does improve encoding performance, it does not obscure which subjects are good or bad. Figure 12 demonstrates subjects with poor encoding performance prior to SRM also tend to have relatively poor (but improved) encoding performance after SRM. This improvement may relate to the number of electrodes in each subject, but this will require further research to fully understand.

---

> ### Author Response · Authors · 2023-11-22
>
> **Question 2** *The comparison of estimated noise ceiling vs. shared responses is more interesting.*
>
> As we describe in detail in response to Weakness 3 and Question 1, we have added a new analysis to the main text of the manuscript in which we quantify how well we can reconstruct shared information in left-out test subjects—and compare this improved noise ceiling to a simpler noise ceiling based on the prior literature.

---

### Meta-Review · Area_Chair_n4hX · 2023-12-15

**Metareview:**

This study aims to investigate whether the shared information space of subjects' brain data can be more accurately predicted than the original brain data using contextual embeddings derived from large language models like GPT-2. The research uses neural responses from eight subjects listening to a podcast, employing a Shared Response Model (SRM) to aggregate data and identify shared stimulus features. Results show significantly improved encoding performance using the SRM-calculated shared space compared to individual brain data.
The combination of utilizing Large Language Models (LLMs) and the Shared Response Model (SRM) to explore common information spaces in human brain processing is a novel approach and the authors show that aligning brains into a shared space significantly improves the encoding performance.
The reviewers raised a couple of concerns and the authors addressed most of them during the author response period and made significant changes/improvements to the paper. The majority of changes improved the clarity of the papers, others incorporated new experiments. The paper is in a much better shape and I would recommend a second round of fresh reviews.

**Justification For Why Not Higher Score:**

None of the reviewers were particularly excited about the paper. The author made significant changes which improved the paper. Nevertheless, it would be good to get a fresh round of reviews.

**Justification For Why Not Lower Score:**

N/A

---

### Decision · Program_Chairs · 2024-01-16

Reject